# DOS-3 mediates cell-non-autonomous DAF-16/FOXO activity in antagonizing age-related loss of *C. elegans* germline stem/progenitor cells

Zhifei Zhang [1], Haiyan Yang[1], Lei Fang[1], Guangrong Zhao [1], Jun Xiang [2] ✉, Jialin C. Zheng [3,4,5,6,7,8,9] ✉ & Zhao Qin [1,7,8] ✉

Age-related depletion of stem cells causes tissue degeneration and failure to tissue regeneration, driving aging at the organismal level. Previously we reported a cell-non-autonomous DAF-16/FOXO activity in antagonizing the age-related loss of germline stem/progenitor cells (GSPCs) in *C. elegans*, indicating that regulation of stem cell aging occurs at the organ system level. Here we discover the molecular effector that links the cell-non-autonomous DAF-16/FOXO activity to GSPC maintenance over time by performing a tissue-specific DAF-16/FOXO transcriptome analysis. Our data show that *dos-3*, which encodes a non-canonical Notch ligand, is a direct transcriptional target of DAF-16/FOXO and mediates the effect of the cell-non-autonomous DAF-16/FOXO activity on GSPC maintenance through activating Notch signaling in the germ line. Importantly, expression of a human homologous protein can functionally substitute for DOS-3 in this scenario. As Notch signaling controls the specification of many tissue stem cells, similar mechanisms may exist in other aging stem cell systems.

Stem cell exhaustion is a hallmark of aging in multicellular organisms[1]. In mammals, age-related decline in stem cell activity affects the normal function and regenerative capacity of many tissues and organ systems including the hematopoietic system, nervous system, muscles, and reproductive system[2]. Failure to maintain stem cells over time is associated with conditions such as tissue degeneration and increased susceptibility to tissue damage, leading to aging at the organismal level. Therefore, alleviating age-dependent depletion of stem cells could be key to achieve healthy aging.

The nematode *C. elegans* provides an attractive model for dissecting the effects of age on stem cell dynamics. Residing at the distal end of the germ line, a pool of proliferating cells comprised of

[1]Key Laboratory of Spine and Spinal Cord Injury Repair and Regeneration of Ministry of Education, Orthopedic Department of Tongji Hospital, School of Medicine, Tongji University, Shanghai 200065, China. [2]Department of Urology, Tongji Hospital, Tongji University School of Medicine, Shanghai 200065, China. [3]Center for Translational Neurodegeneration and Regenerative Therapy, Tongji Hospital Affiliated to Tongji University, Shanghai 200065, China. [4]State Key Laboratory of Cardiology and Medical Innovation Center, Shanghai East Hospital, School of Medicine, Tongji University, Shanghai 200120, China. [5]Shanghai Key Laboratory of Anesthesiology and Brain Functional Modulation, Shanghai Fourth People's Hospital, School of Medicine, Tongji University, Shanghai 200080, China. [6]Translational Research Institute of Brain and Brain-Like Intelligence, Shanghai Fourth People's Hospital affiliated to Tongji University School of Medicine, Shanghai 200080, China. [7]Innovation Center of Medical Basic Research for Brain Aging and Associated Diseases, Ministry of Education, Tongji University, Shanghai 200331, China. [8]Collaborative Innovation Center for Brain Science, Tongji University, Shanghai 200331, China. [9]Shanghai Frontiers Science Center of Nanocatalytic Medicine, Tongji University School of Medicine, Shanghai 200331, China. ✉e-mail: andregene@163.com; jialinzheng@tongji.edu.cn; zqin.med@tongji.edu.cn

germline stem cells and their proliferative progeny (germline stem/progenitor cells, hereafter referred to as GSPCs), support the continuous oogenesis in the adult *C. elegans* hermaphrodites. Self-renewal and differentiation of germline stem cells are controlled by local signals from a single cell, the distal tip cell (DTC). Importantly, the DTC utilizes the highly conserved Notch signaling pathway, a pathway used in many other stem cell paradigms, including mammalian blood, intestine, and epidermal lineages, to promote the

germline stem cell fate[3,4]. Moreover, *C. elegans* is a well-established genetic model for studying the molecular mechanisms of aging—their relatively short (2–3 weeks) lifespan and amenability to genetic manipulations had led to the first identification of many longevity pathways in worms, such as the insulin/IGF-1 signaling (IIS), dietary restriction, and modest inhibition of mitochondrial respiration, all of which were later shown to be highly conserved for aging functions across species[5].

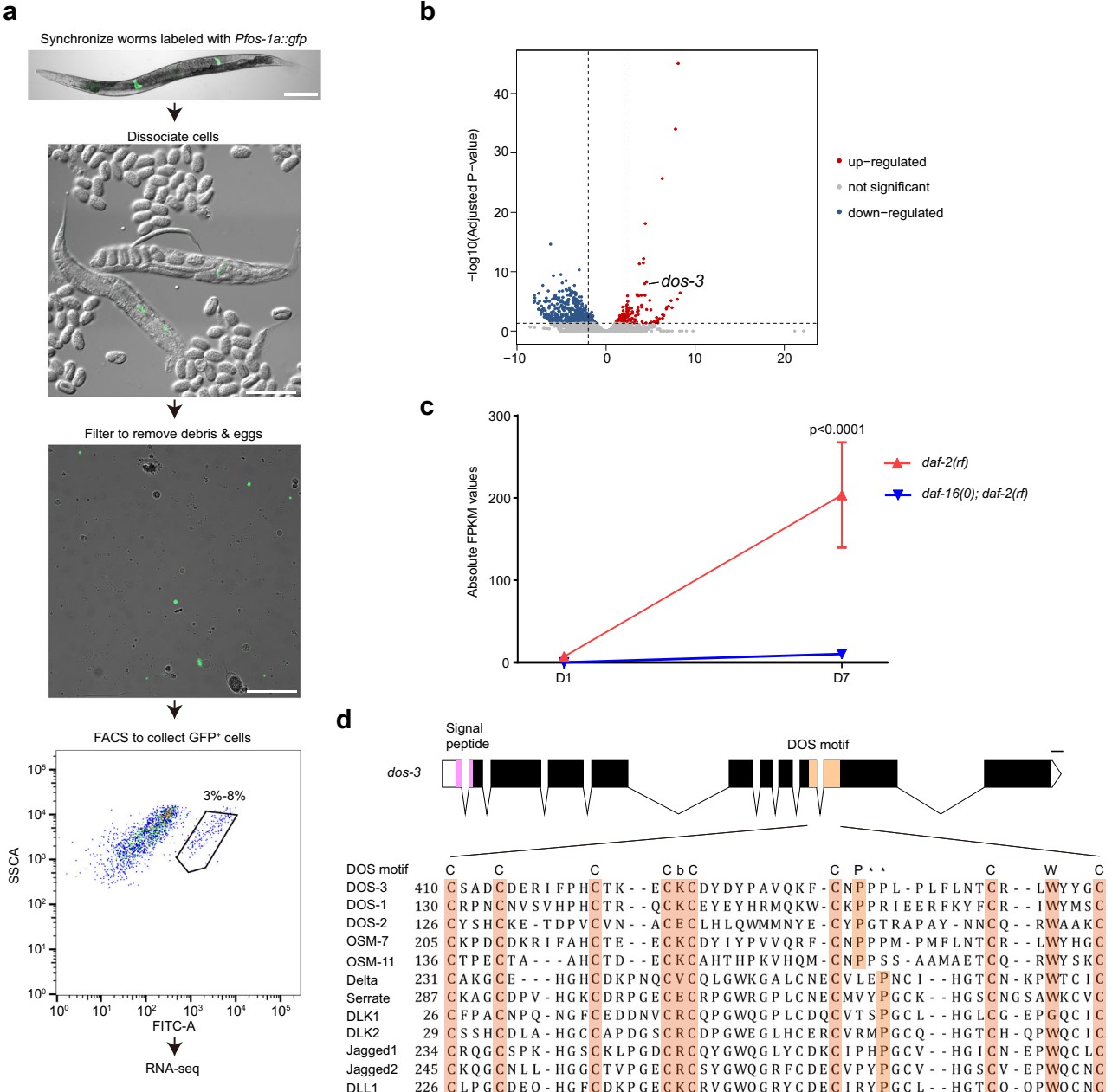

**Fig. 1 | *dos-3*, which encodes a non-canonical Notch ligand, is a DAF-16/FOXO transcriptional target in aging PSG cells. a** Flow chart for RNA-seq transcriptional profiling of isolated adult *C. elegans* PSG cells. Scale bars, 100 µm. **b** Volcano plot comparing mRNAs identified by RNA sequencing of D7 *daf-2(rf)* and *daf-16(0); daf-2(rf)* PSG cells. Red and blue dots represent genes up-regulated and down-regulated in *daf-2(rf)* PSG cells, respectively. **c** RNA-seq data of *dos-3*. Its expression was up-regulated in D7 *daf-2(rf)* PSG. Data are represented as mean ± SEM. *n* = 3 biological replicates. *P* value is calculated using Benjamini-Hochberg test. Source data are provided as a Source Data file. **d** Top: schematic showing the genomic structure of *dos-3*. Graph generated using Exon-Intron Graphic Maker (http://www.wormweb.

org/exonintron). Solid boxes, exons; empty boxes, UTRs; lines, introns. The conserved signal peptide and DOS motif are highlighted in magenta and beige, respectively. Scale bar, 100 bp. Bottom: DOS motif-containing sequences from *C. elegans* (DOS-3, DOS-1, DOS-2, OSM-7, and OSM-11), *Drosophila* (Delta and Serrate), and human (DLK1, DLK2, Jagged1, Jagged2, and DLL1) proteins are aligned under the DOS motif consensus: C-X(3)-C-X(3,8)-C-X(2,5)-C-b-C-X(10,12)-C-X(1,3)-P-X(6,9)-C-X(1,4)-W-X(1,4)-C. Conserved DOS-motif amino acids are shaded. b represents K, V, E, or R, and the asterisks indicate possible positions for proline in the DOS motif.

Previously, we and others reported that similar to stem cells in the adult mammalian nervous system, muscles, skin, and testis[6], the number of *C. elegans* GSPCs decreases dramatically over time[7–11]. The age-associated depletion of GSPCs is regulated by IIS: reduced activity of the sole insulin/IGF-like receptor in *C. elegans*, DAF-2, results in elevated activity of the downstream FOXO transcription factor DAF-16 and consequently delays age-related GSPC loss[8,9]. Surprisingly, we found that DAF-16/FOXO acts in different tissues to maintain GSPCs and to regulate lifespan, and that DAF-16/FOXO is required at the proximal region of the somatic gonad (PSG), specifically in a subset of spermathecal and uterine cells contacting differentiated progeny of GSPCs, to antagonize age-related GSPC loss[9]. These results suggested a novel mechanism of stem cell regulation at the organ system level: in addition to stem cell-niche interaction[3] and regulation of stem cells by systemic environment of the organism[12], cross-talk between stem cells and other non-niche cells within the same organ system may also play an important role in maintaining tissue stem cells[9]. However, how this organ system-level regulation is achieved remained unclear.

In this study, we discover the molecular effector that links PSG DAF-16/FOXO activity to distal GSPC maintenance by performing an adult PSG-specific DAF-16/FOXO transcriptome analysis. *dos-3*, which encodes a non-canonical Notch ligand, was up-regulated by DAF-16/FOXO activity in aging PSG cells of animals with reduced IIS. Promoter analysis suggested that *dos-3* is a direct transcriptional target of DAF-16/FOXO. Functional validation using a null allele of *dos-3* showed that it is required for the effect of reducing IIS on GSPC maintenance over time through activating Notch signaling in the distal germ line. Finally, overexpressing either DOS-3 or a human homologous protein DLK1 could rescue the age-related GSPC loss in wild type worms. By elucidating the mechanism by which cell-non-autonomous DAF-16/FOXO activity antagonizes age-related loss of *C. elegans* GSPCs, our data provide mechanistic insights into the regulation of stem cells by non-niche signals produced by cells within the same organ system. As Notch signaling controls the specification of many tissue stem cells[13], similar mechanisms may exist in other contexts to coordinate the functional output of the relevant organ system.

## Results

### RNA-seq transcriptional profiling to define the DAF-16/FOXO transcriptomes of isolated adult PSG cells

We speculated that a secreted signal, whose expression is controlled by DAF-16/FOXO, mediates the effect of PSG DAF-16/FOXO activity on the distal GSPC pool. To identify relevant DAF-16/FOXO targets, we performed a PSG-specific gene profiling analysis by extracting RNA from isolated PSG cells of a *daf-2*-reduction-of-function (rf) mutant, *daf-2(e1370)* (high DAF-16/FOXO activity) and a *daf-16(0); daf-2(rf)* double mutant with a null allele of *daf-16*, *mu86* (no DAF-16/FOXO activity). To mark cells of the PSG, we obtained a MosSCI insertion that expresses GFP under a PSG-specific promoter[14] (*Pfos-1a::gfp*) and crossed this transgene into the *daf-2(rf)* and *daf-16(0); daf-2(rf)* backgrounds. For cell dissociation, worms were treated with proteinases and mechanically disrupted. After dissociation, GFP⁺ cells were immediately isolated by fluorescence-activated cell sorting (FACS). RNA was extracted from the collected cells and used for RNA-seq experiments (Fig. 1a, Supplementary Fig. 1). The DAF-16/FOXO transcriptomes of isolated young and aging PSG cells were determined by identifying genes that were differentially expressed in the PSG cells of *daf-2(rf)* and *daf-16(0); daf-2(rf)* animals on day 1 (D1) and day 7 (D7) of adulthood, respectively. The complete dataset is available in the Gene Expression Omnibus (GEO) database under accession code GSE243994.

We compared our D1 PSG-specific DAF-16/FOXO transcriptome with the previously reported D1 whole-worm DAF-16/FOXO transcriptome[15,16]. Our dataset was much smaller and although overlap between the two gene lists was observed, most of the PSG DAF-16/

FOXO-regulated genes differed from those identified in the whole-worm set (Supplementary Fig. 2a). This result is consistent with our estimation from the FACS data that PSG cells comprise only 3-8% of the cell population in the adult *C. elegans* hermaphrodites, therefore expression changes of relevant DAF-16/FOXO targets in the PSG could be masked by their expression in other tissues. During the reproductive period of *C. elegans* hermaphrodites, cells in the PSG play an important role in regulating oocyte maturation, ovulation, and fertilization[17,18]. These processes require normal IIS[19]. Consistently, pathway enrichment analysis of our D1 PSG-specific DAF-16/FOXO transcriptome showed that gene clusters associated with these PSG functions such as "calcium ion transmembrane import into cytosol", "striated muscle contraction", and "oogenesis" were overrepresented, confirming that our cell isolation method is highly selective for PSG cells (Supplementary Fig. 2b).

In line with the timing of unmated hermaphrodites' reproductive cessation on ~D5, our D7 PSG-specific DAF-16/FOXO transcriptome was no longer enriched with gene classes associated with PSG functions in reproduction, but with categories involved in stress responses including "mitochondrial unfolded-protein response", "FoxO signaling pathway", as well as "response to heat" (Supplementary Fig. 2c). This up-regulation of stress response genes in the aging *daf-2(rf)* PSG suggests that an anti-aging response is elicited in these cells, which could lead to the production of an intercellular signaling molecule to modulate progenitor activity in the distal germ line.

### Analysis of the aging PSG-specific DAF-16/FOXO transcriptome identifies *dos-3*

While examining our aging PSG-specific DAF-16/FOXO transcriptome, we found a gene *dos-3*, representing a particularly attractive candidate for the signal from PSG to the distal germ line in the *daf-2(rf)* animals for the following reasons: (1) its expression was up-regulated by DAF-16/FOXO activity in the aging *daf-2(rf)* PSG (Fig. 1b, c); (2) it bears a signal peptide sequence at its N-terminus suggestive of a secreted protein (Fig. 1d); (3) it encodes a putative non-canonical Notch ligand. DOS-3 and related *C. elegans* proteins (DOS-1, DOS-2, OSM-7, and OSM-11) do not contain the characteristic DSL domain of canonical Notch ligands, but instead possess a conserved DOS motif that is present in the extracellular domain of many known Notch ligands in other species (Fig. 1d, Supplementary Fig. 3). Although the function of DOS-3 was not characterized, two other DOS proteins, OSM-7 and OSM-11, had been shown to be able to activate LIN-12 and GLP-1, the two Notch receptors in *C. elegans*, to regulate biological processes such as vulval development, adult chemosensory response, and larval molting quiescence[20,21]. We therefore hypothesized that DOS-3 might be the signal from PSG that could act upon the GLP-1/Notch signaling in the distal germ line to affect GSPC maintenance over time.

### *dos-3* is up-regulated in aging *daf-2(rf)* PSG

To define the expression pattern of *dos-3* during aging, we generated two transcriptional reporters of *dos-3* by expressing *gfp* under the control of its endogenous promoter using extrachromosomal arrays (Fig. 2a). We observed similar results for both transgenic lines: GFP fluorescence was detected in the PSG cells of *daf-2(rf)* animals on both D1 and D7 (Fig. 2b, d); additionally, we observed a significant increase in GFP fluorescence intensity in D7 *daf-2(rf)* PSG cells relative to D1, confirming that *dos-3* is up-regulated in aging *daf-2(rf)* PSG (Fig. 2e).

### DAF-16/FOXO binds to *dos-3* promoter

DAF-16/FOXO induces expression of its canonical targets by binding to the DAF-16 binding element (DBE, GTAAAt/cA) of their promoters[15]. To test if *dos-3* is a direct transcriptional target of DAF-16/FOXO, we mutated the two DBEs in the *dos-3* promoter (Fig. 2a) and generated GFP transcriptional reporters using the mutated sequence. For all three transgenic lines, GFP fluorescence was lost in the PSG cells of *daf*-

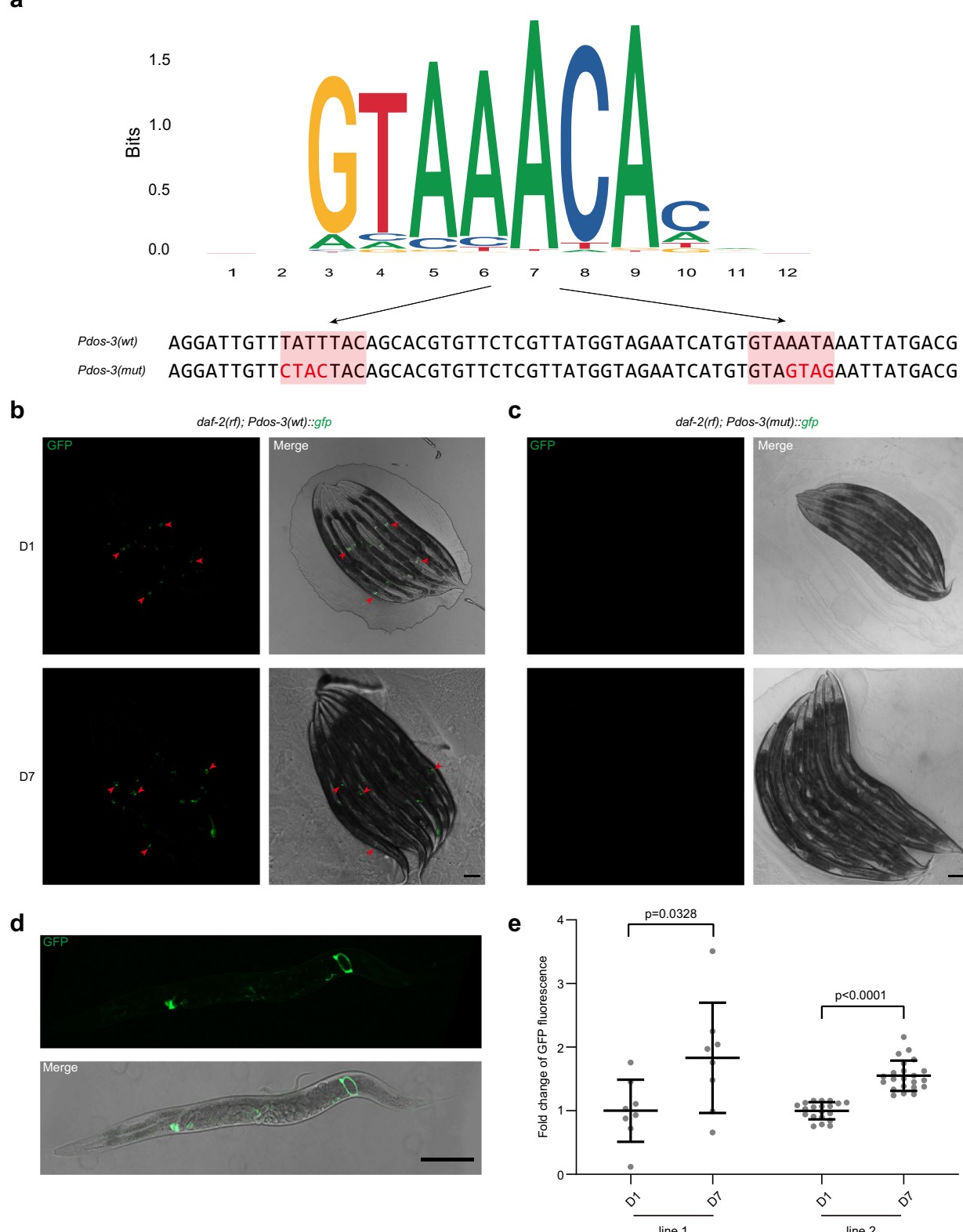

2(rf) animals (Fig. 2c), suggesting that *dos-3* expression requires the binding of DAF-16/FOXO to the DBEs in its promoter and therefore it is a direct target of DAF-16/FOXO.

**Disruption of *dos-3* function promotes GSPC loss in *daf-2(rf)***

To further investigate the function of *dos-3* in GSPC maintenance over time, we generated a loss-of-function allele of *dos-3*, *syb5125*, using the

CRISPR/Cas9 technique. This mutation likely causes a complete loss of function of *dos-3* because it deletes 864 bp after the signal peptide sequence and introduces a single-nucleotide change so that a premature stop codon forms at the deletion site (Fig. 3a).

In order to assess the age-associated depletion of GSPCs, we first measured the number of GSPCs at the beginning of adulthood. As expected, *daf-2(rf)* animals had fewer GSPCs (-140 GSPCs per gonad

**Fig. 2 | *dos-3* is up-regulated in aging *daf-2(rf)* PSG, which requires the binding of DAF-16/FOXO to the two DBEs in its promoter. a** Top: consensus DBE sequences identified from *C. elegans* ChIP-seq data by Jaspar. Bottom: DBE-containing sequences of the *dos-3* promoters used in our transcriptional reporter analysis. The two DBEs are shaded in coral with the mutated bases indicated in red. **b, c** Expression of *dos-3* transcriptional reporters in *daf-2(rf)*. Worms are oriented head toward the upper left corner. Scale bars: 100 μm. **b** Representative GFP and merge images of D1 and D7 *daf-2(rf); Ex[Pdos-3(wt)::gfp]* worms. Arrowheads indicate GFP expression in the PSG cells. **c** Representative GFP and merge images of D1 and D7 *daf-2(rf); Ex[Pdos-3(mut)::gfp]* worms. GFP is undetectable in these animals. **d** PSG expression of *Pdos-3(wt)::gfp* in *daf-2(rf)*. Representative GFP and merge images of an adult *daf-2(rf); Ex[Pdos-3(wt)::gfp]* worm. The worm is oriented head to the left. Scale bar: 100 μm. **e** Fold change of GFP fluorescence in D7 *daf-2(rf); Ex[Pdos-3(wt)::gfp]* worms relative to D1. **e** is a quantification of data represented in (**b**). Two transgenic lines were examined for this experiment. Data are represented as mean ± SD with individual values shown as dots. *n* = 8, 8, 20, 22 animals from left to right. *P* values are calculated using two-tailed Student's *t*-test. Source data are provided as a Source Data file.

arm, Fig. 3c, l) than wild type (~180 GSPCs per gonad arm, Fig. 3b, l) on D1 because IIS promotes larval germline proliferation in *C. elegans*[22]. This phenotype is dependent on DAF-16/FOXO activity[22], therefore loss of *daf-16* reversed the number of GSPCs in *daf-2(rf)* mutants on D1 to a level similar to that in wild type (Fig. 3d, l). By contrast, removing *dos-3* did not alter the number of GSPCs in *daf-2(rf)* animals on D1, suggesting that *dos-3* is not required for the effect of reducing IIS on larval GSPC accumulation (Fig. 3e, l). Lastly, since *C. elegans* uses Notch signaling for their germline stem cell specification, *e2141*, a reduction-of-function mutation of *glp-1*, which encodes the Notch receptor expressed in the germ line, further reduced the number of GSPCs in *daf-2(rf)* mutants on D1 (Fig. 3f, l).

We hypothesized that in aging *daf-2(rf)* animals, DAF-16/FOXO activates expression of DOS-3, which is then secreted from the PSG and acts upon GLP-1 on the distal germ line to maintain GSPCs. As reported previously, we observed a delay in age-related GSPC loss in *daf-2(rf)* mutants[9], shown by an elevation of D7 GSPC number in these animals (~130 GSPCs per gonad arm, Fig. 3h, m) compared with wild type (~110 GSPCs per gonad arm, Fig. 3g, m). Consistent with previous observations, this delay was suppressed by loss of *daf-16*[9], as evidenced by a reduction in D7 GSPC number in *daf-16(0); daf-2(rf)* animals even though they started with more GSPCs on D1 relative to *daf-2(rf)* (Fig. 3d, i, l, m). Importantly, we found that this delay was also suppressed by a mutation in either *dos-3* or *glp-1*. Starting with similar numbers of GSPCs on D1, *dos-3(0) daf-2(rf)* animals maintained significantly fewer GSPCs than *daf-2(rf)* on D7 (Fig. 3e, j, l, m). Likewise, age-related GSPC loss was resumed in *daf-2(rf) glp-1(rf)* worms: these animals had a notably smaller GSPC pool than *daf-2(rf)* on D1, and many became GLP (all germ cells differentiated, no GSPCs remained) with age (Fig. 3f, k, l, m). Together, these results are consistent with our hypothesis that *dos-3* is the main downstream effector of DAF-16/FOXO in regulating GSPC maintenance over time and that it exerts its function by impinging on the same Notch signaling pathway utilized by signals emanating from the DTC.

We used either DAPI nuclear morphology (Fig. 3) or pS/TQ antibody staining which marks early meiotic prophase nuclei[23] (Supplementary Fig. 4) to determine the border of the germline proliferative zone when comparing GSPC numbers of the various genetic groups. Analysis using these two different methods led to the same conclusion: reducing DAF-2 activity delays age-associated depletion of GSPCs and this effect requires *daf-16*, *dos-3*, and *glp-1* (Fig. 3l, m; Supplementary Fig. 4b, c). In addition, we performed phospho-Histone 3 (pH3) antibody staining to measure changes in the GSPC proliferation rates. Although an age-related decrease in mitotic index was detected for all genetic groups, none of the mutations seemed to cause a significant effect on mitotic index (Supplementary Fig. 4d, e). Therefore, we decided to focus on analyzing the number of GSPCs for the rest of this study.

### Adult PSG *dos-3* expression is required for GSPC maintenance over time

We confirmed the adult PSG requirement for *dos-3(+)* in GSPC maintenance over time by tissue-specific *dos-3* RNAi. To achieve adult PSG-specificity, we conducted RNAi knockdown in the *rde-1(0)* RNAi-

deficient animals expressing *rde-1(+)* from the PSG-specific *fos-1a* promoter[24] and fed the worms with RNAi bacteria starting from D1. We found that similar to the case of *daf-16*, knocking down *dos-3* only in the adult PSG resembled knocking down *dos-3* in the whole animal, causing more pronounced GSPC depletion in the *daf-2(rf)* background (Supplementary Fig. 5). This requirement for both *daf-16* and *dos-3* in the adult PSG is in agreement with our hypothesis that transcription factor DAF-16/FOXO activates expression of DOS-3 in the adult PSG cells to antagonize age-related GSPC loss.

### DOS-3 acts upon Notch signaling in the distal germ line

*C. elegans* relies on Notch signaling for germline stem cell specification. The DTC expresses two DSL ligands LAG-2 and APX-1, which interact with GLP-1/Notch receptor on germ cells. Following GLP-1 activation, its intracellular domain gets cleaved and translocates into the nucleus, where it interacts with DNA-binding protein LAG-1 to activate the transcription of downstream Notch effector genes. Two direct transcriptional targets of GLP-1, *sygl-1* and *lst-1*, are redundantly required and each sufficient for promoting the undifferentiated, proliferation-competent germline stem cell fate[25–30]. Previous studies have reported that an age-related decrease in GLP-1/Notch signaling leads to changes in germline stem cell fate specification, contributing to age-related declines in meiotic entry and reproductive output[11]. Conversely, ectopic expression of *sygl-1* could delay reproductive aging and age-related changes in germ cells[31].

To test if DOS-3 exerts its function in GSPC maintenance over time through the previously described Notch signaling pathway in the distal germ line, we examined germline expression of *sygl-1* and *lst-1* in various genetic backgrounds during aging. To visualize SYGL-1 protein, we inserted a 3xOLLAS-encoding sequence into the endogenous *sygl-1* locus using the CRISPR/Cas9 technique to generate an OLLAS::SYGL-1 fusion protein[29] (Fig. 4a). We found that consistent with the age-associated depletion of GSPCs, wildtype animals also displayed an age-related decline in the extent of Notch signaling in the distal germ line as measured by both the number of OLLAS::SYGL-1-expressing cells (Fig. 4b, g, l, m) and the size of OLLAS::SYGL-1-positive domain in cell diameters (Fig. 4b, g, n, o). This result is in line with the observation made by Kocsisova et al. in mated wildtype hermaphrodites[11], suggesting that reduction in Notch signaling contributes to age-related loss of GSPCs regardless of whether the worms are mated or not.

Consistent with changes in their GSPC numbers, the age-related decrease in Notch signaling is greatly attenuated in *daf-2(rf)* worms: even though they started adulthood with fewer OLLAS::SYGL-1-expressing cells and a smaller OLLAS::SYGL-1-positive domain (Fig. 4b, c, l, n), these worms retained more OLLAS::SYGL-1-expressing cells and a larger OLLAS::SYGL-1-positive domain on D7 compared with wild type (Fig. 4g, h, m, o). This result suggests that aging *daf-2(rf)* animals maintain more GSPCs by sustaining a greater extent of Notch signaling in the distal germ line.

To test whether the attenuation of age-related decline in Notch signaling in *daf-2(rf)* is dependent on *daf-16*, *dos-3*, or *glp-1*, we examined OLLAS::SYGL-1 expression in young and aging *daf-16(0); daf-2(rf)*, *dos-3(0) daf-2(rf)*, and *daf-2(rf) glp-1(rf)* worms. We found that although removing *daf-16* or *dos-3* had no effect on the extent of Notch signaling

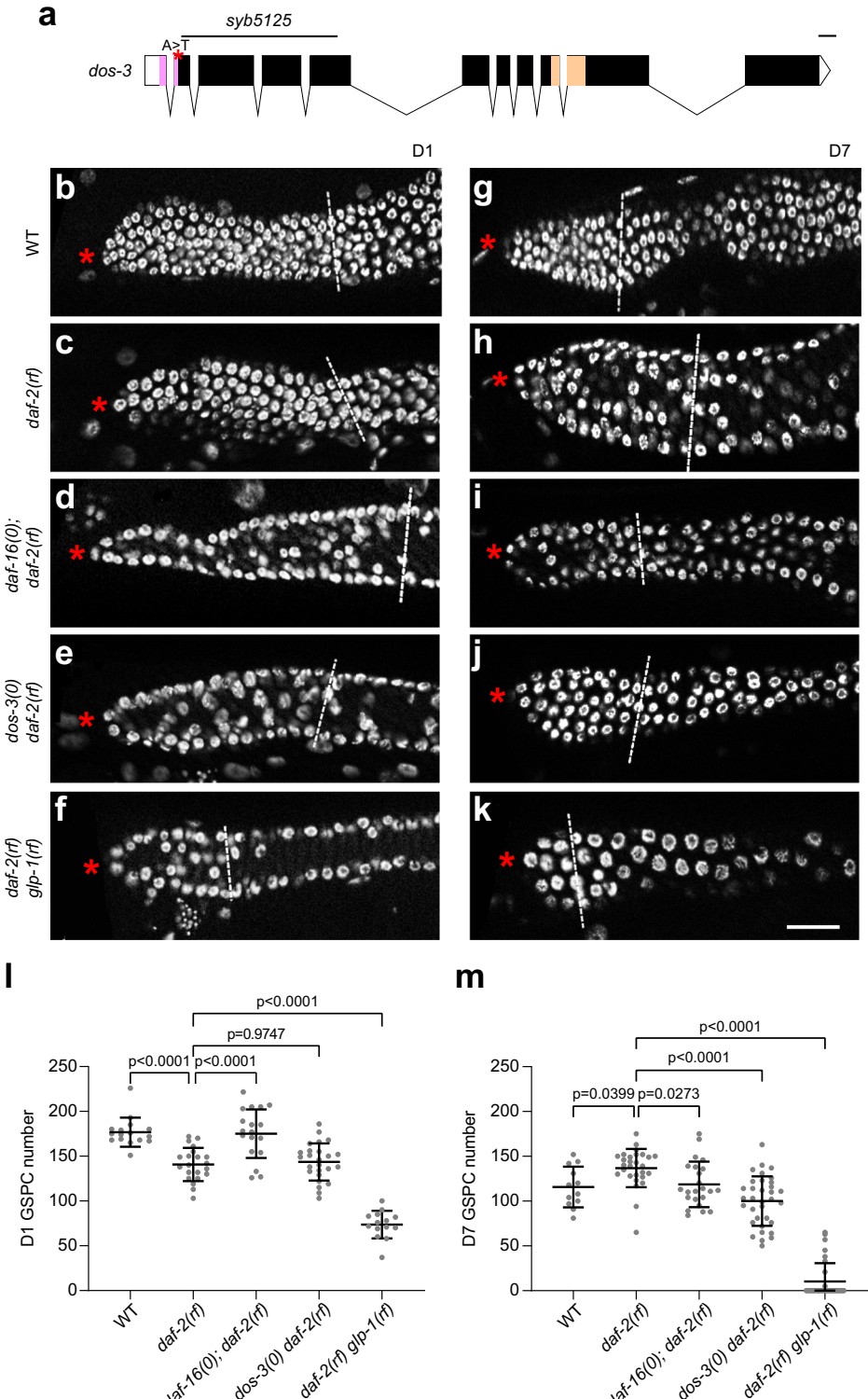

**Fig. 3 | dos-3 is required for the effect of reducing IIS on GSPC maintenance over time in C. elegans. a** Schematic showing the *dos-3(syb5125)* mutation. The deletion and A to T single-nucleotide change in the *dos-3(syb5125)* allele are indicated by black line and red asterisk, respectively. Graph generated using Exon-Intron Graphic Maker (http://www.wormweb.org/exonintron). Scale bar, 100 bp. **b**–**k** DAPI-stained germ lines of D1 and D7 adult worms. Asterisk indicates the distal end of the germ line, and the white dashed line indicates the proximal border of the proliferative zone, defined as the first row of cells in which two or more crescent-shaped meiotic prophase nuclei appear. Scale bar: 20 μm. Representative germ lines of D1 wildtype (**b**), *daf-2(rf)* (**c**), *daf-16(0); daf-2(rf)* (**d**), *dos-3(0) daf-2(rf)* (**e**), and *daf-2(rf) glp-1(rf)* (**f**) animals. Representative germ lines of D7 wildtype (**g**), *daf-2(rf)* (**h**), *daf-16(0); daf-2(rf)* (**i**), *dos-3(0) daf-2(rf)* (**j**), and *daf-2(rf) glp-1(rf)* (**k**) animals. Number of GSPCs per gonad arm in D1 (**l**) and D7 (**m**) wildtype, *daf-2(rf)*, *daf-16(0); daf-2(rf)*, *dos-3(0) daf-2(rf)*, and *daf-2(rf) glp-1(rf)* worms. Data are represented as mean ± SD with individual values shown as dots. *n* = 16, 21, 18, 25, 14 animals from left to right in (**l**); *n* = 12, 28, 23, 34, 32 animals in (**m**). *P* values are calculated using one-way ANOVA and Dunnett's post hoc test. Source data are provided as a Source Data file.

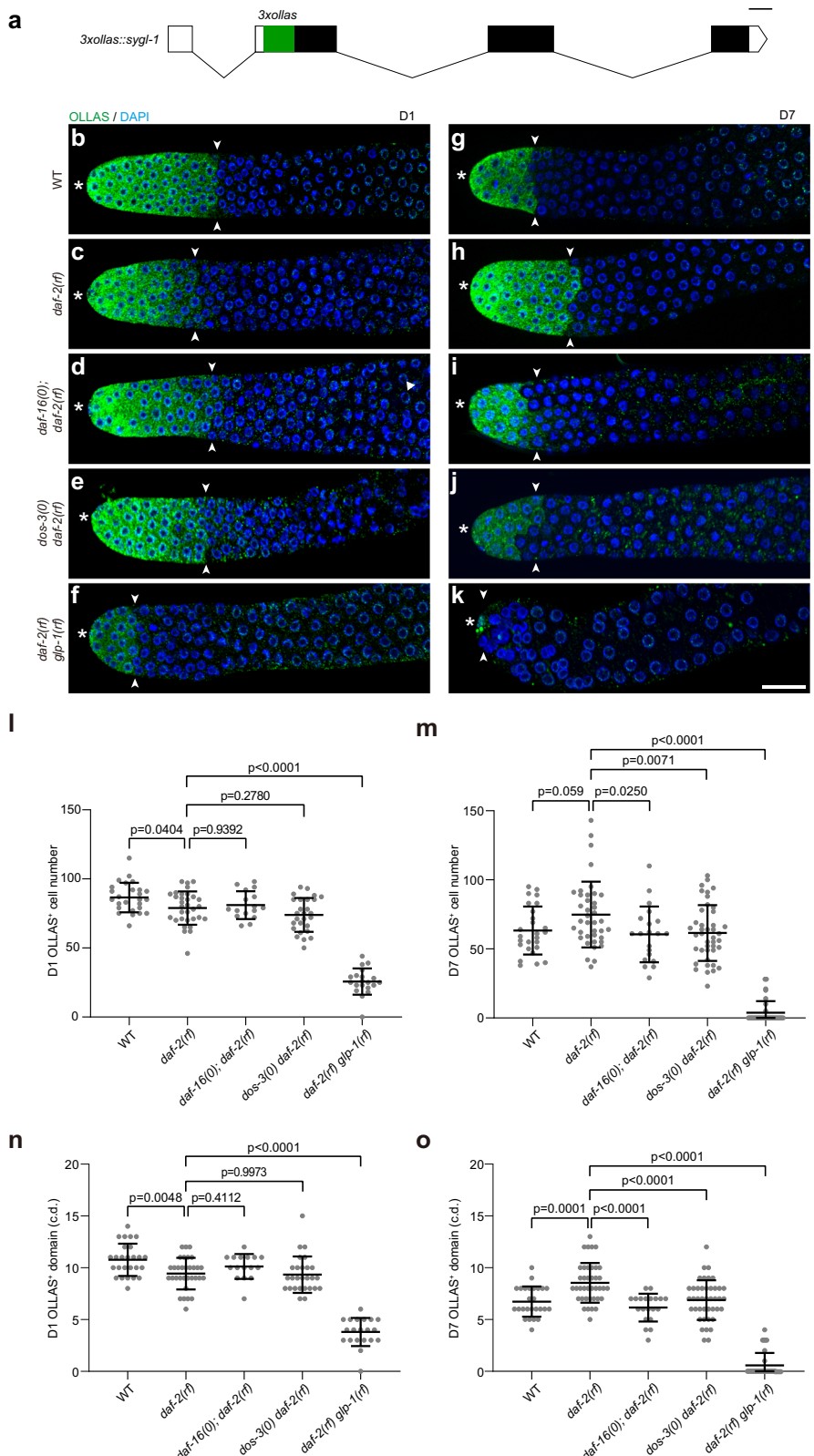

in the distal germ line of *daf-2(rf)* animals at the beginning of adulthood (Fig. 4c–e, l, n), both significantly reduced the number of OLLAS::SYGL-1-expressing cells and the size of OLLAS::SYGL-1-positive domain on D7 (Fig. 4h–j, m, o). Consistent with being a direct transcriptional target of GLP-1/Notch, OLLAS::SYGL-1 expression was dramatically decreased in *daf-2(rf) glp-1(rf)* worms compared with *daf-2(rf)* at both time points (Fig. 4c, f, h, k). Notably, the extent of Notch signaling in the distal

germ line of *daf-2(rf) glp-1(rf)* animals was further reduced on D7 relative to D1 (Fig. 4f, k), suggesting that the attenuation of age-related decline in Notch signaling caused by reducing IIS also requires *glp-1*.

We examined germline expression of *lst-1* by generating a LST-1::OLLAS fusion protein[29] (Supplementary Fig. 6a), and found that changes in *sygl-1* and *lst-1* expression in the various genetic backgrounds during aging show excellent agreement (Supplementary

**Fig. 4 | DOS-3 activates Notch signaling in the distal germ line of aging *daf-2(rf)* animals. a** Schematic showing the CRISPR/Cas9 knock-in of *3xollas* at the N-terminus of *sygl-1* endogenous locus. Graph generated using Exon-Intron Graphic Maker (http://www.wormweb.org/exonintron). Scale bar, 100 bp. **b**–**k** Dissected gonads from D1 and D7 adults stained with anti-OLLAS antibody (green) and DAPI (blue). Asterisk indicates the distal end of the germ line, and the white arrowheads indicate the proximal border of the OLLAS staining. Scale bar: 20 µm. Representative germ lines of D1 wildtype (**b**), *daf-2(rf)* (**c**), *daf-16(0); daf-2(rf)* (**d**), *dos-3(0) daf-2(rf)* (**e**), and *daf-2(rf) glp-1(rf)* (**f**) animals expressing OLLAS::SYGL-1. Representative germ lines of D7 wildtype (**g**), *daf-2(rf)* (**h**), *daf-16(0); daf-2(rf)* (**i**), *dos-3(0)*

*daf-2(rf)* (**j**), and *daf-2(rf) glp-1(rf)* (**k**) animals expressing OLLAS::SYGL-1. Number of OLLAS+ cells per gonad arm in D1 (**l**) and D7 (**m**) wildtype, *daf-2(rf)*, *daf-16(0); daf-2(rf)*, *dos-3(0) daf-2(rf)*, and *daf-2(rf) glp-1(rf)* worms expressing OLLAS::SYGL-1. Size of OLLAS+ domain measured in cell diameters (c.d.) in D1 (**n**) and D7 (**o**) wildtype, *daf-2(rf)*, *daf-16(0); daf-2(rf)*, *dos-3(0) daf-2(rf)*, and *daf-2(rf) glp-1(rf)* worms expressing OLLAS::SYGL-1. Data are represented as mean ± SD with individual values shown as dots. $n = 26, 32, 16, 27, 20$ animals from left to right in (**l**, **n**); $n = 26, 39, 20, 42, 33$ animals in (**m**, **o**). *P* values are calculated using one-way ANOVA and Dunnett's post hoc test. Source data are provided as a Source Data file.

Fig. 6). Together, these data correlate well with our GSPC counts shown in Fig. 3, and are consistent with our hypothesis that DOS-3 mediates the effect of reducing IIS on GSPC maintenance over time by activating Notch signaling in the distal germ line.

### *dos-3* overexpression rescues age-related GSPC loss

Since *dos-3* is required in *daf-2(rf)* animals for better maintenance of GSPCs, we went on to test if overexpressing *dos-3* could alleviate age-related GSPC loss in wild type. We generated extrachromosomal arrays expressing *dos-3* cDNA under the control of a heat shock promoter, *Phsp-16.2*. Worms with the transgene and their non-transgenic siblings were either maintained at 20 °C or subjected to daily 1-h heat shock at 35 °C from D1 to D6. We found that heat shock induction of both transgenes was able to increase the number of D7 GSPCs (Fig. 5a). Therefore, we conclude that *dos-3* overexpression is sufficient to improve GSPC maintenance with age.

Our hypothesis predicts that *dos-3* overexpression would compensate for loss of *daf-16*. We therefore tested whether ectopic *dos-3* expression could rescue age-related GSPC loss in the *daf-16(0); daf-2(rf)* mutant. Indeed, heat shock induction of the same *dos-3*-expressing transgenes increased D7 GSPC counts of *daf-16(0); daf-2(rf)* animals (Fig. 5b), consistent with *dos-3* being downstream of *daf-16* in the pathway.

Based on the existing literature, it is unclear if the DOS-only proteins act as proper Notch ligands by themselves or if they function as co-ligands that facilitate the action of canonical DSL family Notch ligands. To differentiate these two possibilities, we examined the dependence of the effect of *dos-3* overexpression on the presence of *lag-2*, which encodes one of the two DSL ligands expressed by the DTC. We found that *dos-3* overexpression rescued age-associated loss of GSPCs even in a *lag-2(rf)* background, *lag-2(q420)* (Fig. 5c). On the contrary, *dos-3* overexpression was not able to improve GSPC maintenance over time in the *glp-1(rf)* background, confirming that *dos-3* acts via *glp-1* (Fig. 5d). Lastly, ectopic *dos-3* expression alone elevated distal germline expression of both Notch targets, *sygl-1* and *lst-1*, on D7 (Fig. 5e, f). Together, these data suggest that DOS-3 could function as a proper Notch ligand by itself, which acts upon GLP-1/Notch receptor and activates downstream Notch targets in the distal germ line.

### Human DLK1 can substitute for DOS-3

To test if this function of a *C. elegans* non-canonical Notch ligand is conserved across species, we repeated the above rescue experiment using a human DOS protein, DLK1. Indeed, we found that expression of a soluble isoform of DLK1[32] could functionally substitute for DOS-3 in vivo in *C. elegans*, increasing the number of GSPCs on D7 in wildtype worms (Fig. 5g). This result is consistent with DOS proteins exerting their functions by activating Notch signaling and suggests that the underlying molecular mechanism may be conserved during evolution.

### Discussion

Our data show that *dos-3*, a direct transcriptional target of DAF-16/FOXO, encodes a non-canonical Notch ligand that mediates the effect of reducing IIS on GSPC maintenance over time through activating Notch signaling in the distal germ line (Fig. 6). Like other tissue stem

cells that maintain the homeostasis of their organ systems, changes in *C. elegans* GSPCs impact greatly on the functional output of the reproductive system, and age-related loss of GSPCs is a key driver of *C. elegans* reproductive cessation[33]. Furthermore, *C. elegans* germ line employs the highly conserved Notch signaling pathway to specify stem cell fate as in many other stem cell systems such as the neural, hematopoietic, intestinal, and epidermal lineages in mammals[13]. Therefore, our results offer two important implications relevant to stem cell aging in general.

First, our study elucidates a mechanism by which regulation of stem cell aging is achieved at the organ system level. Our previous analysis of age-related loss of GSPCs in *C. elegans* suggested that modulation at the organ system level is present in this stem cell system[9]. In this study, our data show that by producing a secreted signal, cells at the proximal region of the worm's reproductive tract can influence the behavior of GSPCs at the distal end. Similar mechanisms are expected to operate in other stem cell paradigms to coordinate the activity of different cells within the same organ system for maintaining its proper function.

Second, our results imply a function of Notch signaling in stem cell aging. Our data show that the secreted signal that mediates the effect of reducing IIS on GSPC maintenance over time is a soluble Notch ligand which increases Notch signaling in the distal germ line. Importantly, overexpressing either the *C. elegans* Notch ligand or a human homologous protein could rescue age-related GSPC loss in wildtype animals. These results are consistent with the observations made by Kocsisova et al. that in wildtype worms, the level of Notch signaling in the distal germ line decreases with age, as evidenced by the diminished sizes of *sygl-1* and *lst-1*-expressing domains[11] and that sustaining the activity of Notch signaling by ectopic expression of *sygl-1* in the germ line of aging animals could delay age-related declines in proliferative zone output and progeny production[31]. As Notch signaling is similarly required for the specification of many other tissue stem cells, we speculate that it may be involved in the aging of those cells as well.

In addition to providing those important implications relevant to stem cell aging in general as discussed above, our finding also highlights the utility of *C. elegans* as a tool for dissecting the intricate regulation of Notch signaling. The invariant lineage of *C. elegans* vulval development helped identification of *lin-12/Notch* in the 1980s which initiated the use of *C. elegans* as a model for understanding the genetics of Notch signaling[34]. Since then, many genetic screens have been carried out in *C. elegans* to identify core components of the signaling pathway and to elucidate the mechanism of signal transduction[25]. Here, our data showcase the importance of using *C. elegans* as a model to appreciate the complexity of Notch ligands. Unlike Notch ligands in other commonly used genetic models such as *Drosophila* and zebrafish which contain both DSL domain and DOS motif, the mammalian and *C. elegans* genomes encode several non-canonical Notch ligands that only have the DOS motif but lack the DSL domain (Supplementary Fig. 3a). The biological function of these DOS motif-only ligands and their modes of action are poorly understood. Although the mammalian DOS protein DLK1 has been implicated in organism development and tissue regeneration through both Notch-

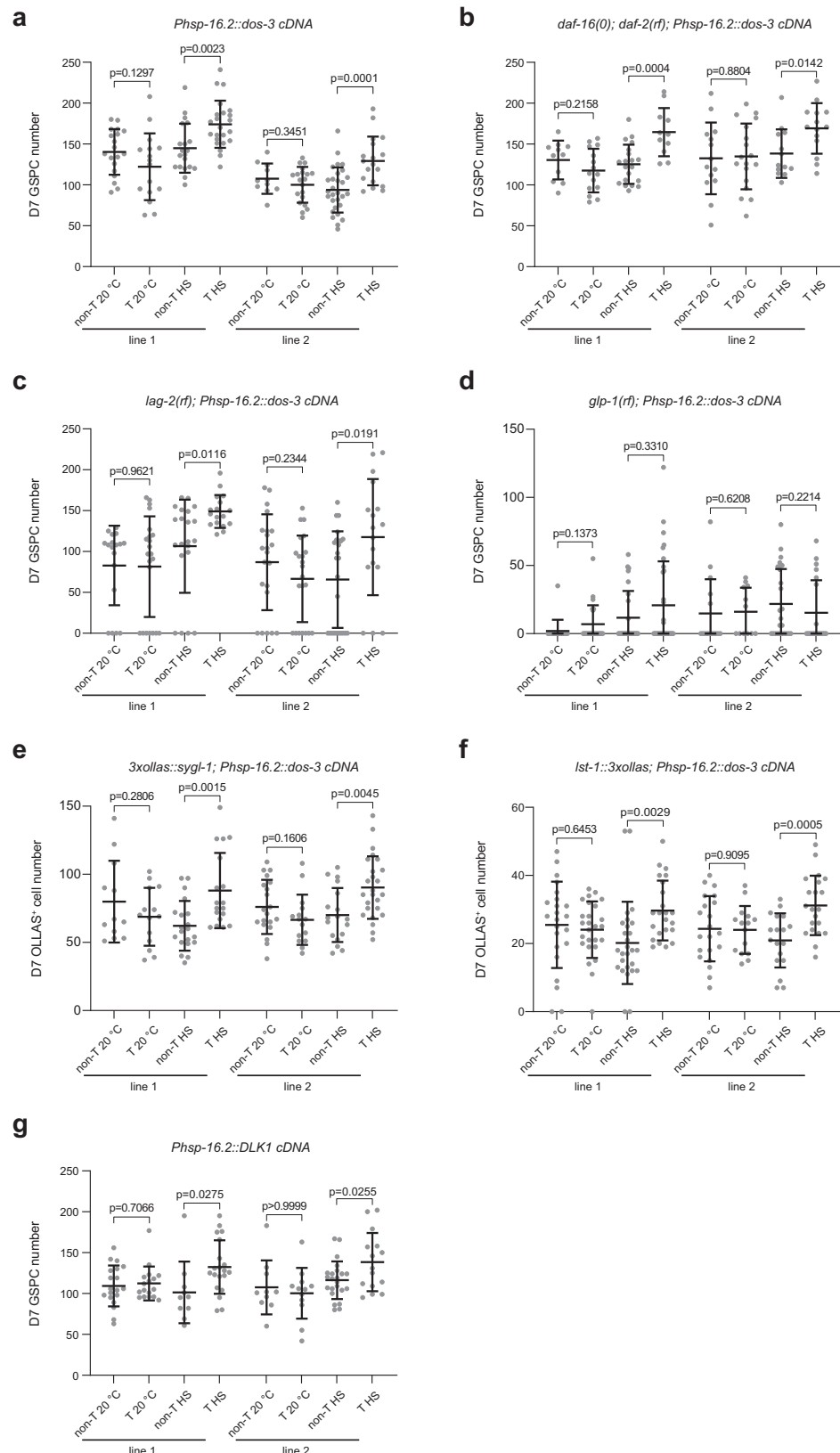

dependent and -independent mechanisms[35], our results show for the first time that the *C. elegans* DOS protein DOS-3 influences stem cell aging, presumably by functioning as a soluble Notch ligand which could contribute to the complex and precise regulation of signaling. Moreover, our data provide evidence for DOS-only proteins to act as proper Notch ligands by themselves rather than co-ligands that require activity of other DSL ligands to activate downstream Notch signaling.

Our previous analysis showed that GSPCs in *C. elegans* are used up as reproduction progresses, and that blocking germ cell flux through the reproductive tract maintains GSPCs partly through PSG DAF-16/ FOXO activity[9]. Based on results from this study, we speculate that

**Fig. 5 | Overexpression of *dos-3* or human *DLK1* rescues age-related GSPC loss.**
**a** Number of GSPCs per gonad arm in D7 *Ex(Phsp-16.2::dos-3 cDNA)* worms and their non-transgenic siblings with or without daily heat shock treatment. **b** Number of GSPCs per gonad arm in D7 *daf-16(0); daf-2(rf); Ex(Phsp-16.2::dos-3 cDNA)* worms and their non-transgenic siblings with or without daily heat shock treatment. **c** Number of GSPCs per gonad arm in D7 *lag-2(rf); Ex(Phsp-16.2::dos-3 cDNA)* worms and their non-transgenic siblings with or without daily heat shock treatment. **d** Number of GSPCs per gonad arm in D7 *glp-1(rf); Ex(Phsp-16.2::dos-3 cDNA)* worms and their non-transgenic siblings with or without daily heat shock treatment. **e** Number of OLLAS⁺ cells per gonad arm in D7 *Ex(Phsp-16.2::dos-3 cDNA)* worms and their non-transgenic siblings expressing OLLAS::SYGL-1 with or without daily heat shock treatment.
**f** Number of OLLAS⁺ cells per gonad arm in D7 *Ex(Phsp-16.2::dos-3 cDNA)* worms and

their non-transgenic siblings expressing LST-1::OLLAS with or without daily heat shock treatment. **g** Number of GSPCs per gonad arm in D7 *Ex(Phsp-16.2::DLK1 cDNA)* worms and their non-transgenic siblings with or without daily heat shock treatment. Two transgenic lines were examined for all experiments. non-T non-transgenic, T transgenic, HS heat shock. Data are represented as mean ± SD with individual values shown as dots. $n = 19, 16, 20, 23, 11, 20, 29, 18$ animals from left to right in (**a**); $n = 11, 15, 20, 11, 14, 18, 13, 14$ animals in (**b**); $n = 18, 22, 22, 17, 22, 22, 25, 17$ animals in (**c**); $n = 18, 23, 26, 32, 15, 12, 24, 20$ animals in (**d**); $n = 12, 14, 20, 18, 20, 15, 18, 25$ animals in (**e**); $n = 23, 26, 27, 23, 21, 15, 18, 21$ animals in (**f**); $n = 19, 17, 10, 20, 10, 12, 23, 15$ animals in (**g**). *P* values are calculated using two-tailed Student's *t*-test. Source data are provided as a Source Data file.

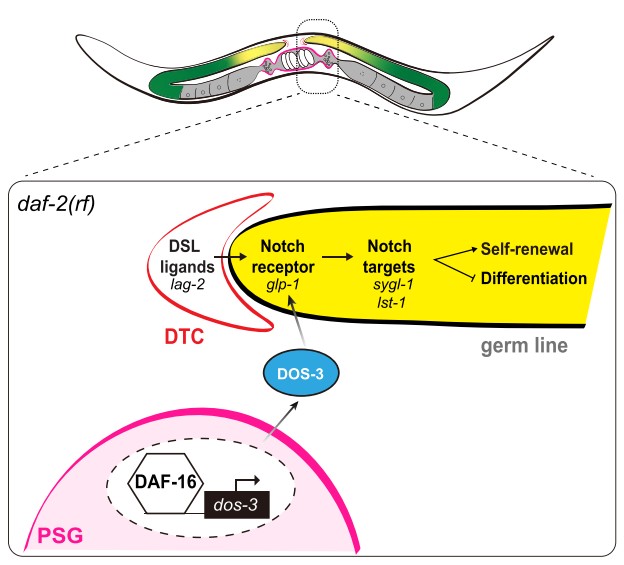

**Fig. 6 | Model.** In aging *daf-2(rf)* animals, DAF-16/FOXO activates expression of DOS-3, which is then secreted from the PSG and acts upon GLP-1/Notch receptor on the distal germ line to maintain GSPCs through the downstream Notch signaling. Parts of the worm gonad are indicated by a color scheme: red, DTC; magenta, PSG; yellow, GSPCs; green, differentiating germ cells.

conditions of reduced germ cell flux may inhibit IIS in the PSG cells, which in turn induces DOS-3 expression by activating DAF-16/FOXO. As a result, DOS-3 may partially account for the effect of reducing germ cell flux on GSPC maintenance. Additional interesting questions such as the exact links between reducing germ cell flux and DAF-16/FOXO activation in the PSG cells, and the mechanism by which germ cell flux influences GSPC maintenance that is independent of PSG DAF-16/FOXO activity remain for future inquiry.

## Methods

### *C. elegans* strains and maintenance

Strains were derived from N2 wild type (Bristol) and handled using standard methods[36]. Unless otherwise indicated, worms were grown on nematode growth medium (NGM) plates seeded with OP50 at 20 °C. All strains used in this study are listed in Supplementary Table 1.

### Generation of *C. elegans* strains

The PSG *gfp*-expressing transgene, *dos-3* mutant allele, and *ollas* knock-in animals were generated at SunyBiotech (Fuzhou, China). To generate the PSG *gfp*-expressing transgene, a 7.4 kb *fos-1a* promoter, a *gfp*-encoding sequence, and a 728 bp *unc-54* 3'UTR were cloned into the pCFJ350 vector using the ClonExpress MultiS One Step Cloning Kit (Vazyme, cat#: C113).

For transcriptional reporters of *dos-3*, the 1 kb endogenous *dos-3* promoter was amplified from N2 genomic DNA (primers: 5'-

GATTCTTGGGAACTCGTCGTTAA-3' and 5'-ACAGATCGCAACA-GAGTGTG-3') and a DBE-mutated *dos-3* promoter was subsequently generated using the Mut Express II Fast Mutagenesis Kit V2 (Vazyme, cat#: C214). Each promoter was cloned into the pPD95.75 Fire vector to generate the *Pdos-3(wt)::gfp* and *Pdos-3(mut)::gfp* plasmids, respectively. The resulting constructs were then injected to N2 at a concentration of 20 ng/μL together with 10 ng/μL of *Pmyo-3::mcherry* as a co-injection marker. Two or three independent transgenic lines were generated and maintained as extrachromosomal arrays.

For heat-shock rescuing constructs of *dos-3* and *DLK1*, the *dos-3*-encoding sequence was amplified from *daf-2(rf)* cDNA (primers: 5'-ATGAAGTCACTAGTATTTTATTTTTGG-3' and 5'-TTAATCA-TAATCATCAGGAATATTAAGGAAAGT-3') and the human *DLK1*-encoding sequence was synthesized (Sangon Biotech). A 400 bp *hsp-16.2* promoter along with either *dos-3* or *DLK1* cDNA were cloned into the pPD95.75 Fire vector. The resulting constructs were then injected to N2 at a concentration of 20 ng/μL together with 20 ng/μL of *Pmyo-2::gfp* as a co-injection marker. Two independent transgenic lines were generated and maintained as extrachromosomal arrays for both constructs.

Standard crossing procedure was used to generate double mutants and to put the transgenes into different mutant backgrounds. All worm strains generated in this study are available from Dr. Zhao Qin upon request.

### FACS isolation of adult PSG cells

Adult PSG cell isolation was performed using a method adapted from the protocol described in Kaletsky et al., 2016 for isolating adult *C. elegans* neurons[37]. Worms labeled with *Pfos-1a::gfp* were synchronized by hypochlorite treatment. To obtain a large population of synchronized young and aging adults, synchronized L1 larvae were grown on peptone-enriched 8P plates seeded with OP50 until D1 and D7 of adulthood, respectively. For D7 samples, worms were transferred onto fresh plates every day after washing them off plates with M9 buffer and filtering with 40 μm strainers to discard eggs and larvae. For each cell dissociation, ~20,000 animals were collected and washed five times with M9 to remove the bacteria food. The worms were washed one more time with 500 μL lysis buffer (200 mM DTT, 0.25% SDS, 20 mM HEPES pH 8.0, and 3% sucrose) and incubated in 750 μL lysis buffer at room temperature for 6.5 min. They were then washed five times with M9 and incubated in 250 μL of freshly made 20 mg/mL Pronase (Sigma-Aldrich, cat#: P8811) for 20 min at room temperature. During the incubation, the worm suspension was vigorously pipetted 100 times every 5 min to release cells. After the last session of pipetting, 750 μL PBS was added to the worm suspension and the mixture was centrifuged at 350 *g* for 5 min. Supernatant was discarded, and the pellet was resuspended in 500 μL Accutase (Gibco, cat#: A6964) with 20 strokes of delicate pipetting. The cell suspension was then incubated at room temperature for up to 10 min. A 2 μL sample was examined under a Zeiss Imager M1 using an AF488 filter (Carl Zeiss) to assess the level of dissociation. If sufficient level of dissociation was achieved, the cell suspension was centrifuged at 350 *g* for 5 min. Supernatant was

discarded, and the pellet was resuspended in ice-cold PBS supplemented with 2% FBS (Gibco, cat#: 30044333). The dissociated cell suspension was then passed through a 40 μm filter (Falcon, cat#: 352340) to remove eggs and debris. Cells were sorted using a FACSArialI cell sorter (BD Biosciences), and gating parameters (cell size and fluorescence intensity) were set by reference to a control sample of dissociated, unlabeled adult wildtype worm cells. 10,000-200,000 GFP+ cells were collected into Trizol (Ambion, cat#: 15596026).

## RNA isolation, amplification, library preparation, and sequencing

RNA of isolated D1 and D7 PSG cells was extracted using a standard Trizol/chloroform/isopropanol method, and RNA integrity was assessed using the Bioanalyzer 2100 system (Agilent Technologies). RNA samples were sequenced and analyzed by Novogene (Beijing, China). Three biological replicates were performed for *daf-2(rf)* and *daf-16(0); daf-2(rf)* PSG cells at both time points. Specifically, 10–400 ng of total RNA with a RIN value > 6.0 was used as input for the SMARTer Ultra Low RNA Kit for Illumina Sequencing (Takara, cat#: 634936) to generate DNA fragments ranging from 200-500 bp suitable for Illumina sequencing[38]. Sequencing was conducted on an Illumina Novaseq 6000 platform, yielding 150 bp paired-end reads.

## RNA-seq data analysis

Raw data (raw reads) were processed through FASTQC, and all downstream analyses were based on clean data with high quality. The paired-end clean reads were aligned to the *C. elegans* genome (WBcel235) using Hisat2 v2.0.5. Mapped reads of each gene were counted using featureCounts v1.5.0-p3, and genes with more than 10 reads in all replicates were defined as "expressed".

Differential expression analysis was performed using the DESeq2 R package (1.20.0), and the resulting *P* values were adjusted using the Benjamini and Hochberg's approach for controlling false discovery rate. FPKM (Fragments Per Kilobase of exon model per Million mapped fragments) value of each gene was calculated and used for Pearson correlation coefficient analysis and principal component analysis by scikit-learn[39]. Genes with an adjusted *P* < 0.05 & |log2(FoldChange)| ≥ 1 were assigned as "differentially expressed". Heatmaps were generated in R using pheatmap (v.1.0.12) from FPKM values, and Venn diagrams were created with venerable (v.3.1.0.9000).

Pathway enrichment analysis of differentially expressed genes was performed using Metascape[40] with the whole genome as the background list. For each gene list, up- and down-regulated genes were analyzed separately. Top 10 enriched gene clusters were presented.

## Protein sequence alignment and phylogenetic analysis

The following protein sequences are aligned in Supplementary Fig. 3, identified by species, isoform, and NCBI reference sequence code: OSM-7 (*Caenorhabditis elegans*, isoform a, NP_001255209.1), OSM-11 (*Caenorhabditis elegans*, NP_510823.1), DOS-1 (*Caenorhabditis elegans*, NP_499017.2), DOS-2 (*Caenorhabditis elegans*, NP_494703.2), DOS-3 (*Caenorhabditis elegans*, NP_497271.3), LAG-2 (*Caenorhabditis elegans*, NP_503877.1), APX-1 (*Caenorhabditis elegans*, isoform a, NP_001360546.1), ARG-1 (*Caenorhabditis elegans*, NP_001024615.1), DSL-1 (*Caenorhabditis elegans*, NP_500054.1), DSL-2 (*Caenorhabditis elegans*, NP_500052.1), DSL-3 (*Caenorhabditis elegans*, isoform a, NP_500108.1), DSL-4 (*Caenorhabditis elegans*, NP_001359594.1), DSL-5 (*Caenorhabditis elegans*, NP_502191.1), DSL-6 (*Caenorhabditis elegans*, NP_502148.2), DSL-7 (*Caenorhabditis elegans*, NP_001023803.1), Delta (*Drosophila melanogaster*, isoform a, NP_477264.1), Serrate (*Drosophila melanogaster*, isoform a, NP_524527.3), C901 (*Drosophila melanogaster*, NP_572673.1), DeltaA (*Danio rerio*, NP_571029.2), DeltaB (*Danio rerio*, NP_571033.1), DeltaC (*Danio rerio*, NP_571019.1), DeltaD (*Danio rerio*, NP_571030.2), DLK1 (*Homo sapiens*, isoform 1, NP_003827.4), DLK2 (*Homo sapiens*, isoform a, NP_996262.1), DLL1

(*Homo sapiens*, NP_005609.3), DLL3 (*Homo sapiens*, isoform 1, NP_058637.1), DLL4 (*Homo sapiens*, NP_061947.1), Jagged1 (*Homo sapiens*, NP_000205.1), Jagged2 (*Homo sapiens*, isoform a, NP_002217.3). The alignment was performed using MUSCLE. The best protein model was determined as "WAG model and Gamma distributed(G)" by using the MODEL package in MEGA 11. The phylogenetic tree was then constructed with the WAG + G protein model using the maximum likelihood tree method.

## GFP microscopy of *dos-3* transcriptional reporters

Synchronized worms were examined under a Zeiss LSM880 confocal laser scanning microscope (Carl Zeiss). The PSG expression of *Pdos-3(wt)::gfp* in every D1 and D7 *daf-2(rf)* worm analyzed was confirmed by first locating the PSG using DIC and then checking the GFP channel. Images of D1 and D7 animals were acquired using the same setting including exposure time. The PSG region of each worm was outlined and fluorescence intensity within the area was obtained using the Zeiss ZEN2.3 pro software. Data were normalized to the average fluorescence intensity of D1 samples for each line for comparison.

## RNAi

RNAi by bacterial feeding was performed as described[41]: worms were synchronized by egg-laying and grown on NGM plates seeded with OP50. On D1, worms were transferred to plates seeded with HT115 carrying L4440 (control), *daf-16*, or *dos-3* RNAi plasmid until the end of the experiment. To generate the *dos-3* RNAi construct, a 455 bp *dos-3* cDNA sequence was amplified from *daf-2(rf)* cDNA (primers: 5′-GTATCGATAAGCTTGATCAAATGTGACTACGATTACCC-3′ and 5′-AGACCGGCAGATCTGATCGCCGCAATTAGATGAAC-3′) and cloned into the L4440 empty vector using the ClonExpress MultiS One Step Cloning Kit (Vazyme, cat#: C113).

## Gonad dissection and immunohistochemistry

Gonad dissection and immunohistochemistry were performed using a method adapted from protocols described previously[11,29]. Briefly, worms were washed with PBS and transferred onto a dissecting watch glass. To immobilize the worms, levamisole was used at a final concentration of 200 μM. The dissection was carried out using a pair of 25 G 5/8″ needles, with incisions usually made at the tail. Dissected gonads were transferred to low-retention centrifuge tubes using glass pipettes, and subjected to a fixation procedure consisting of incubation first in 3% paraformaldehyde (PFA) for 10 min at room temperature and then in 100% methanol at −20 °C for a minimum of 2 h. Following fixation, the gonads were washed three times with PBS + 0.1% Tween-20 (PBST). They were then incubated with the primary antibody at 4 °C overnight, washed again with PBST three times, and subsequently incubated with the secondary antibody at room temperature for 2 h. The stained gonads were washed with PBST three times and suspended in a drop of VECTASHIELD mounting medium with DAPI (Vector Laboratories, cat#: H-1200). Fluorescent images of the distal germ line were captured within 72 h using a Zeiss Imager M1 equipped with an Apotome Axioimager (Carl Zeiss).

Primary antibodies used in this study included rabbit anti-pS/TQ (1:500, Cell Signaling Technology, cat#: 6966, lot#: 8), mouse anti-pH3 (1:150, Cell Signaling Technology, cat#: 9706, lot#: 10), and rat anti-OLLAS (1:2000, Novus Biologicals, cat#: NBP1-06713, lot#: F16). Secondary antibodies used in this study included Alexa 488 goat anti-rabbit IgG (1:600, Jackson, cat#: 111-545-003, lot#: 168597), Alexa 594 goat anti-mouse IgG (1:600, Jackson, cat#: 115-585-003, lot#: 168330), and Alexa 488 donkey anti-rat IgG (1:500, Invitrogen, cat#: A-21208, lot#: A21208).

## Analysis of GSPCs

The number of GSPCs included all the germ cells between the distal tip and the beginning of meiotic entry, which was defined either by DAPI

nuclear morphology[9] or by pS/TQ antibody staining[23]. For DAPI, worms were fixed in 100% ethanol for 5–7 min depending on their age and stained using VECTASHIELD mounting medium with DAPI (Vector Laboratories, cat#: H-1200). Z-stack images of the distal germ line were acquired using a Zeiss Imager M1 equipped with an Apotome Axioimager (Carl Zeiss). The border between the proliferative zone and the transition zone was marked as the first row of cells in which two or more nuclei displayed the characteristic crescent shape. For pS/TQ, gonad dissection and immunohistochemistry were performed as described above. The proximal border of the proliferative zone was determined as the first row of cells in which at least half of the nuclei were pS/TQ$^+$.

Mitotic index of GSPCs was calculated as the percentage of pH3$^+$ nuclei over the total number of proliferative zone nuclei.

## Estimation of *sygl-1-* or *lst-1*-expressing cells in the distal germ line

Quantification of *sygl-1-* or *lst-1*-expressing cells was enabled by the usage of strains expressing the 3xOLLAS epitope tag from the endogenous *sygl-1* or *lst-1* locus. Distal germ cells displaying a cytoplasmic OLLAS signal surpassing background level were defined as "OLLAS$^+$". The size of OLLAS$^+$ domain was determined in cell diameters from the distal tip to the last row with half or more OLLAS$^+$ cells.

## DOS-3 and DLK1 rescue

Worms were synchronized by egg-laying and grown at 20 °C until D1 of adulthood. Heat shock was done by immersing parafilm wrapped plates in a pre-heated water bath at 35 °C for 1 h every day for 6 days. Worms were examined on D7.

## Statistics and reproducibility

Data are representative of at least three biological replicates with similar results. Details of statistical analyses are reported in the figure legends. Statistical analyses were performed using GraphPad Prism and Adobe Illustrator was used to make figures.

## Reporting summary

Further information on research design is available in the Nature Portfolio Reporting Summary linked to this article.

## Data availability

The RNA-seq data generated in this study have been deposited in the Gene Expression Omnibus (GEO) database under accession code GSE243994. Source data are provided with this paper.

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

## Acknowledgements
We thank Jane Hubbard, Zhiyong Shao, Haijun Tu, Tim Schedl, and Theodora Tolkin for reagents and/or advice. This work was supported by the National Natural Science Foundation of China (81771503), Shanghai Rising-Star Program (19QA1409700), Tongji Hospital Start-up Funding for Scientific Research (RCQD2301) to Z.Q., the State Key Program of the National Natural Science Foundation of China (81830037), the Major Research Plan of the National Natural Science Foundation of China (91949204) to J.C.Z., and the Shanghai Municipal Health Commission's Research Project (202040078) to J.X. Some strains were provided by the CGC, which is funded by NIH Office of Research Infrastructure Programs (P40 OD010440).

## Author contributions
Conceptualization, Z.Q.; Methodology, Z.Q.; Investigation, Z.Z., H.Y., L.F., and G.Z.; Writing—Original Draft, Z.Q. and Z.Z.; Writing—Review & Editing, Z.Q. and Z.Z.; Funding Acquisition, Z.Q., J.C.Z., and J.X.; Resources, Z.Q., J.C.Z., and J.X.; Supervision, Z.Q.

## Competing interests
The authors declare no competing interests.
