## [Peer Review File · Nature Communications]

DOS-3 mediates cell-non-autonomous DAF-16/FOXO activity in antagonizing age-related loss of *C. elegans* germline stem/progenitor cellsReviewer #1 (Remarks to the Author):

Review of the manuscript by Z. Zhang et al.

The depletion of stem cells is thought to be one of the main causes for tissue degeneration during ageing. The progressive loss of germline stem cells/progenitor cells (GSPC) in the mitotic zone of the *C. elegans* gonads has been a good model to study the mechanisms underlying stem cell depletion during ageing.

This manuscript is a continuation of the previously published work by Qin and colleagues (2015), which reported that insulin signaling in the soma, specifically in the proximal somatic gonad, promotes stem cell loss by repressing the DAF-16 FOXO transcription factor. Here, Zhang and colleagues describe a mechanism, through which DAF-16 in the proximal somatic gonad maintains the pool of GSPCs in the distal gonad. By determining the transcriptome of purified proximal somatic gonad cells they identify the non-canonical NOTCH ligand DOS-3 as a direct DAF-16 target that may be secreted by the PSG to activate the GLP-1 NOTCH receptor in the GSPCs to sustain their proliferation.

Overall, this is a clean-cut story that certainly advances our current knowledge. How general this mechanism of cross-tissue NOTCH signaling via a soluble ligand is in stem cell maintenance is difficult to judge based on the data in this manuscript. I think, this point is pushed a little too hard in the discussion (lines 296ff).

I have two major points of criticism:

(1) Based on the existing literature, it is unclear if the DOS-only proteins (OSM-5 and OSM-11 being the first members of this family) act as proper NOTCH ligands by themselves or if they function as co-ligands that facilitate the action of canonical DSL family NOTCH ligands. This question could be tested here. For example, is the effect of DOS-3 (or DLK) overexpression dependent on the presence of LAG-2 or APX-1, i.e. can DOS-3 overexpression rescue the loss of GSPCs even in *lag-2(lf)* mutants? Moreover, does ectopic DOS-3 expression alone change the expression of the GLP-1 targets (e.g. *lst-1* & *sygl-1*) and does it rescue the *daf-2(lf)*; *daf-16(lf)* phenotype? Epistasis analysis with *dos-3 lf* and overexpression combined with *glp-1 gf* or *lf* mutants could be used to test if *dos-3* acts via *glp-1*, as proposed.

(2) There is a technical issue: Most of the data depend on the accurate quantification of GSPC numbers, and some of the effects are rather small (approximately 20% change in GSPC number). The authors manually draw lines between the mitotic and transition zone based on nuclear morphology visualized by DAPI staining (line 480: "...the beginning of meiotic entry, defined as the first row of cells in which two or more nuclei displayed the characteristic crescent shape."). I strongly recommend to use the more accurate and unbiased methods that are commonly used in the field, such as HIM-3 antibody staining to mark the start of the transition zone, and staining mitotic cells with phospho H3 antibodies for an accurate quantification of the mitotic index to measure possible changes in the GSPC proliferation rates.

Specific comments:

- 1) To the cell isolation (Fig. 1a): What is the enrichment of the isolated cells? This can be estimated from the FACS data (i.e. the gating of the negative controls)- which should be shown as supplementary information) or from microscopic inspection of the sorted cells.
- 2) The clustering of the transcriptomic data seems odd (extended Fig 1 b&c): The samples neither cluster by the genotype nor by age. Especially, the D7 *daf-2*; *daf-16* samples are closest to the D1 *daf-2* samples, although they show the opposite effect in GSPC numbers. This should be commented.
- 3) The *dos-3>gfp* reporter expression in Fig 2b: the magnification of the images is far too low. It is impossible to tell if *dos-3* is expressed in the PSG as stated (line 147) or in other tissues. How was fluorescence quantified and how were intensities calibrated? (not explained in the methods)
- 4) Fig. 3: the wild-type controls should be shown. Without them it is impossible to follow their line of arguments (lines 181f)
- 5) Numbers of animals scored are missing in all graphs (dots are difficult to count)
- 6) Statistics: how were t-test corrected for multiple comparisons?

Reviewer #2 (Remarks to the Author):

Review of "A non-canonical Notch ligand mediates cell-non-autonomous DAF-16/FOXO antagonizing age-related loss of *C. elegans* germline stem/progenitor cells" by Zhang et al.

This report identifies *dos-3*, encoding a non-canonical Notch ligand, as a direct target of the transcription factor DAF-16/FOXO in the proximal somatic gonad (PSG). Moreover, *DOS-3* expression in the PSG promotes positively regulates germline stem and progenitor cell proliferation in the distal germline, apparently via the GLP-1/Notch receptor. *DOS-3* activity limits stem cell loss in the aging *C. elegans* germ line, and the human *DOS-3* homolog can rescue loss of *C. elegans* *DOS-3* activity. Previously, it was unclear how DAF-16/FOXO activity in the PSG regulates GSPCs in aging animals in a cell-non-autonomous manner. Here, the authors used RNA-seq to identify targets of DAF-16/FOXO activity in the PSG and focused their analysis of *das-3*. The authors provide evidence that *dos-3* is a transcriptional target of DAF-16/FOXO, necessary and sufficient for GSPC maintenance, and acts via the Notch receptor, GLP-1, promote GSPC proliferation. This work sheds considerable light on the relationship among insulin/IGF-1 signaling, the aging germline, and Notch signaling, and it will be interest to many readers.

Comments

1) Line 59 and following: Add some detail to this section to provide the reader with a better understanding of previous work and to better describe the rationale for your work.

It would help to state some relationships in a straightforward way. For example, "Reduced activity of the sole insulin/IGF-like receptor in *C. elegans*, DAF-2, results in elevated DAF-16 activity and consequently delays age-related GSPC loss." [Bold used only for emphasis.]

Clarify "anatomically separable," for example "Surprisingly, we found that DAF-16/FOXO acts in different cells/tissues to maintain GSPCs and regulate lifespan, and that ..." [Bold used only for emphasis.]

Introduce the term "germ cell flux" in the Introduction. It's currently used only in the Discussion.

2) The first time a genetic mutation is mentioned, provide some information about the specific allele that was used and tell the reader if it is a partial loss-of-function allele or a null allele. Also, define "-" and "rf." Typically, *C. elegans* researchers use "0" to designate a null allele.

3) There are many places in the paper where a reference should be provided. For example, include a reference for PSG-specific expression of *fos-1a* (line 93), which is critical to interpreting the RNAi data.

What is the reference for saying proximal somatic gonad cells comprise 5% of somatic cells in the adult herm?

4) Line 171 and Fig. 4: Why isn't wildtype data shown in the figure?

5) Line 185 and Fig. 3: Again, why isn't the wildtype data shown in parts b and f of the figure?

6) Lines 200-205: Clarify the description of the experiment comparing global *daf-16* and *dos-3* knockdown with knockdown only in the PSGs. For example, reword the sentence to read "...knocking down *dos-3* only in the adult PSGs resembled knocking down *dos-3* in the whole animal, ..." [Bold used only for emphasis.]

7) Line 229: Substitute "whether" for "no matter"

8) Line 259 and following: Consider including this section with the earlier section starting on line 196 in order to combine the required and sufficient experiments. Fig. 5a could be added to Fig. 3.

9) Line 279-280: Substitute "that maintain" for "to" so the line reads "Like other tissue stem cells that maintain the homeostasis of their organ systems, ..."

10) Line 324: I think the authors mean to say "...influences stem cell aging, presumably by functioning as a soluble Notch ligand..."

11) Figure 1 legend: Describe the volcano plot in terms of the two datasets that are being compared. For example, "b. Volcano plot comparing mRNAs identified by RNA sequencing of D7 ..."

12) Figure 2 legend: Clearly state that panel d is a quantification of data represented in panel b.

13) Figure 3. Part b-e: State how the proliferative zone was identified (parts b-e). By DAPI morphology?

Part f: Has every pairwise comparison been tested for significance? The authors state that *daf-16(-);daf-2(rf)* is not different from *dos-3(-);daf-2(rf)*. However, by eye the difference between those two genotypes looks very similar to the difference between *daf-2(rf)* and *dpy-16(-);daf-2(rf)*, which is indicated as sig different by Student's t-test.

14) Figure 5 legend: Expand on the model description. The illustration suggests DOS-3 is secreted into the pseudocoelom and binds GLP-1 expressed on the distal germ cell plasma membranes. A schematic illustration of the gonad arm might be helpful here.

15) Line 438 and extended data Fig. 2: State in the legend (or elsewhere) what background list of genes used in the enrichment analysis. Was it the whole genome or a particular subset?

We thank the reviewers for their insightful comments on our manuscript entitled “A non-canonical Notch ligand mediates cell-non-autonomous DAF-16/FOXO activity in antagonizing age-related loss of *C. elegans* germline stem/progenitor cells”. In their comments, both reviewers raised interesting points and we are happy to address them in our point-by-point response below (reviewer comments in blue and our responses in black; relevant changes in manuscript text highlighted in yellow):

Reviewer #1 (Remarks to the Author):

Review of the manuscript by Z. Zhang et al.

The depletion of stem cells is thought to be one of the main causes for tissue degeneration during ageing. The progressive loss of germline stem cells/progenitor cells (GSPC) in the mitotic zone of the *C. elegans* gonads has been a good model to study the mechanisms underlying stem cell depletion during ageing.

This manuscript is a continuation of the previously published work by Qin and colleagues (2015), which reported that insulin signaling in the soma, specifically in the proximal somatic gonad, promotes stem cell loss by repressing the DAF-16 FOXO transcription factor. Here, Zhang and colleagues describe a mechanism, through which DAF-16 in the proximal somatic gonad maintains the pool of GSPCs in the distal gonad. By determining the transcriptome of purified proximal somatic gonad cells they identify the non-canonical NOTCH ligand DOS-3 as a direct DAF-16 target that may be secreted by the PSG to activate the GLP-1 NOTCH receptor in the GSPCs to sustain their proliferation.

Overall, this is a clean-cut story that certainly advances our current knowledge. How general this mechanism of cross-tissue NOTCH signaling via a soluble ligand is in stem cell maintenance is difficult to judge based on the data in this manuscript. I think, this point is pushed a little too hard in the discussion (lines 296ff).

We thank the reviewer’s overall assessment of our work as “a clean-cut story that certainly advances our current knowledge”. We are excited to see from our data that cross-tissue Notch signaling via a soluble ligand influences GSPC maintenance in *C. elegans*, and we agree with the reviewer that how general this mechanism is in other stem cell systems needs to be further tested. Therefore, we have toned down our argument in the discussion by stating that “As Notch signaling is similarly required for the specification of many other tissue stem cells, we speculate that it may be involved in the aging of those cells as well.”

I have to major points of criticism:

(1) Based on the existing literature, it is unclear if the DOS-only proteins (OSM-5 and OSM-11 being the first members of this family) act as proper NOTCH ligands by themselves or if they function as co-ligands that facilitate the action of canonical DSL family NOTCH ligands. This question could be tested here. For example, is the effect of DOS-3 (or DLK) overexpression dependent on the presence of LAG-2 or APX-1, i.e. can DOS-3 overexpression rescue the loss of GSPCs even in lag-2(lf) mutants? Moreover, does ectopic DOS-3 expression alone change the expression of the GLP-1 targets (e.g. lst-1 & sygl-1) and does it rescue the daf-2(lf); daf-16(lf) phenotype? Epistasis analysis with dos-3 lf and overexpression combined with glp-1 gf or lf mutants could be used to test if dos-3 acts via glp-1, as proposed.

We took the reviewer’s suggestion and examined

(1) the effect of *dos-3* overexpression in a *lag-2(rf)* mutant, *lag-2(q420)*. We found that ectopic *dos-3* expression was able to rescue age-associated loss of GSPCs even in *lag-2(rf)* animals. Please see new Fig. 5c and related text in the results section “*dos-3* overexpression rescues age-related GSPC loss”.

(2) the effect of *dos-3* overexpression on the expression of GLP-1 targets, *sygl-1* and *lst-1*. We found that ectopic *dos-3* expression alone elevated distal germline expression of both Notch targets on D7. The data are now in Fig. 5e, f and in the results section “*dos-3* overexpression rescues age-related GSPC loss”.

(3) the effect of *dos-3* overexpression in *daf-16(0); daf-2(rf)* mutants. We found that overexpression of *dos-3* increased D7 GSPC counts of *daf-16(0); daf-2(rf)* animals, consistent with *dos-3* being downstream of *daf-16* in the pathway. Please see new Fig. 5b and related text in the results section “*dos-3* overexpression rescues age-related GSPC loss”.

(4) the effect of *dos-3* overexpression in *glp-1(rf)* mutants. We found that *dos-3* overexpression was not able to improve GSPC maintenance over time in the *glp-1(rf)* background, confirming that *dos-3* acts via *glp-1*. Please see new Fig. 5d and related text in the results section “*dos-3* overexpression rescues age-related GSPC loss”.

Together, these data are in favor of DOS-3 acting as a proper Notch ligand by itself rather than functioning as a co-ligand that facilitates the action of canonical DSL family Notch ligands and are in agreement with our model that DOS-3 mediates the effect of DAF-16 activity on GSPC maintenance over time through activating Notch signaling in the distal germ line.

(2) There is a technical issue: Most of the data depend on the accurate quantification of GSPC numbers, and some of the effects are rather small (approximately 20% change in GSPC number). The authors manually draw lines between the mitotic and transition zone based on nuclear morphology visualized by DAPI staining (line 480: “...the beginning of meiotic entry, defined as the first row of cells in which two or more nuclei displayed the characteristic crescent shape.”). I strongly recommend to use the more accurate and unbiased methods that are commonly used in the field, such as HIM-3 antibody staining to mark the start of the transition zone, and staining mitotic cells with phospho H3 antibodies for an accurate quantification of the mitotic index to measure possible changes in the GSPC proliferation rates.

We thank the reviewer for this comment. As the reviewer pointed out, researchers in the *C. elegans* germ cell field use REC-8/HIM-3 antibody staining or DAPI nuclear morphology to mark the border between the proliferative zone and the transition zone (Hubbard and Schedl, 2019). We chose to use the latter to analyze aging GSPCs is because unlike REC-8/HIM-3 antibody staining, our DAPI staining protocol does not require gonad dissection. Based on our observations and as reported by other groups such as in Kocsisova et al., 2022 bioRxiv, dissected gonads from old worms are fragile, thus causing an unintentional bias in the results of antibody staining toward gonads from less fragile animals in an aging population. Therefore, we used nuclear morphology visualized by DAPI staining to mark the proximal boundary of the proliferative zone in our analysis of aging GSPCs in Qin and Hubbard, 2015. This manuscript is a follow-up of the previous study and we keep the method consistent.

Discrepancies in the exact GSPC numbers in aging worm populations were summarized and discussed in Kocsisova et al., 2019 and in Tolkin and Hubbard, 2021. Nonetheless, studies from different laboratories using different markers to delineate the proximal boarder of the proliferative zone confirmed that the number of GSPCs reduces over time.

For the reason discussed above, we measure mitotic index of aging *C. elegans* germ lines by calculating the percentage of metaphase and anaphase figures identified by DAPI staining over the total number of proliferative zone nuclei rather than by performing phospho H3 antibody staining and calculating the percentage of phospho H3-positive nuclei. In our previous study (Qin and Hubbard, 2015), we reported that the mitotic index of D12 wild type (~0.5%) was reduced relative to D1 (~1.4%). In the current analysis, however, we feel it is difficult to compare the mitotic index of the various genetic groups due to the low proliferation rate of aging GSPCs (see the figure below) and we decide to focus on the number of GSPCs, which is a more robust parameter.

Specific comments:

1) To the cell isolation (Fig. 1a): What is the enrichment of the isolated cells? This can be estimated from the FACS data (i.e. the gating of the negative controls)- which should be shown as supplementary information) or from microscopic inspection of the sorted cells.

We thank the reviewer for this comment. To isolate GFP-labeled PSG cells, we set FACS gating parameters (cell size and fluorescence intensity) by reference to a control sample of dissociated, unlabeled adult wildtype worm cells. For samples from worms labeled with *Pfos-1a::gfp*, we collected GFP⁺ cells in the boxed area shown on the flow cytometry scatter plot in Fig. 1a, which comprised 3-8% of the entire population. This information is now added to the revised Fig. 1a and the methods section “FACS isolation of adult PSG cells”.

2) The clustering of the transcriptomic data seems odd (extended Fig 1 b&c): The samples neither cluster by the genotype nor by age. Especially, the D7 daf-2; daf-16 samples are closest to the D1 daf-2 samples, although they show the opposite effect in GSPC numbers. This should be commented.

We agree with the reviewer that the clustering of our transcriptomic data doesn't seem intuitive: the expression profiles of isolated PSG cells from young and aging *daf-2* and *daf-16; daf-2*

animals neither cluster by genotype nor by age. We think this is probably because from D1 to D7, *C. elegans* hermaphrodites experience dramatic decline in reproductive activities such as oocyte maturation, ovulation, and fertilization. These activities involve PSG cells (Kim, Spike, and Greenstein, 2013; Marcello, Singaravelu, and Singson, 2013) and require normal insulin signaling (Lopez et al., 2013 and our unpublished results). Therefore, it is not surprising to see that PSG samples of the same genotype (but at different ages) or at the same age (but with varying DAF-16 levels) don't cluster together.

3) The *dos-3>gfp* reporter expression in Fig 2b: the magnification of the images is far too low. It is impossible to tell if *dos-3* is expressed in the PSG as stated (line 147) or in other tissues. How was fluorescence quantified and how were intensities calibrated? (not explained in the methods)

We thank the reviewer for this comment. We confirmed the PSG expression of *Pdos-3(wt)::gfp* in every D1 and D7 *daf-2(rf)* worm analyzed by first locating the PSG using DIC and then checking the GFP channel. Images of D1 and D7 animals were acquired using the same setting including exposure time. The PSG region of each worm was outlined and fluorescence intensity within the area was obtained using the Zeiss ZEN2.3 pro software. Data were normalized to the average fluorescence intensity of D1 samples for each line for comparison. This information is now added to the methods section "GFP microscopy of *dos-3* transcriptional reporters". We have also added a higher magnification image of an individual *daf-2(rf); Pdos-3(wt)::gfp* worm as the new Fig. 2d to show the PSG expression of the transgene.

4) Fig. 3: the wild-type controls should be shown. Without them it is impossible to follow their line of arguments (lines 181f)

We took the reviewer's suggestion and added the wildtype data to the new Fig. 3.

5) Numbers of animals scored are missing in all graphs (dots are difficult to count)

We took the reviewer's suggestion and added the numbers of animals scored to the relevant figure legends in the revised manuscript.

6) Statistics: how were t-test corrected for multiple comparisons?

We thank the reviewer for this comment. In our revised manuscript, we use one-way ANOVA and LSD post hoc test for comparisons involving 3 samples (Fig. S4) and use one-way ANOVA and Dunnett's post hoc test for comparisons involving more than 3 samples (Fig. 3, 4 and Fig. S5).

Reviewer #2 (Remarks to the Author):

Review of "A non-canonical Notch ligand mediates cell-non-autonomous DAF-16/FOXO antagonizing age-related loss of *C. elegans* germline stem/progenitor cells" by Zhang et al. This report identifies *dos-3*, encoding a non-canonical Notch ligand, as a direct target of the transcription factor DAF-16/FOXO in the proximal somatic gonad (PSG). Moreover, *DOS-3* expression in the PSG positively regulates germline stem and progenitor cell proliferation in the distal germline, apparently via the GLP-1/Notch receptor. *DOS-3* activity limits stem cell loss in the aging *C. elegans* germ line, and the human *DOS-3* homolog can rescue loss of *C. elegans* *DOS-3* activity. Previously, it was unclear how DAF-16/FOXO activity in the PSG regulates GSPCs in aging animals in a cell-non-autonomous manner. Here, the authors used RNA-

seq to identify targets of DAF-16/FOXO activity in the PSG and focused their analysis of *das-3*. The authors provide evidence that *dos-3* is a transcriptional target of DAF-16/FOXO, necessary and sufficient for GSPC maintenance, and acts via the Notch receptor, GLP-1, promote GSPC proliferation. This work sheds considerable light on the relationship among insulin/IGF-1 signaling, the aging germline, and Notch signaling, and it will be interest to many readers. Thank you!

Comments

1) Line 59 and following: Add some detail to this section to provide the reader with a better understanding of previous work and to better describe the rational for your work.

We took the reviewer's suggestion and added more details to this section.

It would help to state some relationships in a straightforward way. For example, "Reduced activity of the sole insulin/IGF-like receptor in *C. elegans*, DAF-2, results in elevated DAF-16 activity and consequently delays age-related GSPC loss." [Bold used only for emphasis.]

We took the reviewer's suggestion and changed the wording.

Clarify "anatomically separable," for example "Surprisingly, we found that DAF-16/FOXO acts in different cells/tissues to maintain GSPCs and regulate lifespan, and that ..." [Bold used only for emphasis.]

We took the reviewer's suggestion and made the change.

Introduce the germ "germ cell flux" in the Introduction. It's currently used only in the Discussion.

We thank the reviewer for this comment. We would like to focus this study on the intriguing phenotype of *daf-2(rf)* worms and on the mechanism by which reducing IIS maintains more GSPCs than wild type during aging. In our previous analysis (Qin and Hubbard, 2015), we showed that blocking germ cell flux through the reproductive tract also maintains GSPCs. However, the underlying mechanism is not discussed in this manuscript. For these reasons, we prefer to just touch upon the effect of germ cell flux on GSPC maintenance over time in the discussion and save it for future analysis.

2) The first time a genetic mutation is mentioned, provide some information about the specific allele that was used and tell the reader if it is a partial loss-of-function allele or a null allele. Also, define "-" and "rf." Typically, *C. elegans* researchers use "0" to designate a null allele.

We thank the reviewer for this comment. We now provide information about the specific allele that was used in the revised text where a genetic mutation is mentioned for the first time. We use "reduction-of-function (rf)" to indicate a partial loss-of-function allele and "0" to designate a null allele.

3) There are many places in the paper where a reference should be provided. For example, include a reference for PSG-specific expression of *fos-1a* (line 93), which is critical to interpreting the RNAi data.

We took the reviewer's suggestion and added relevant references that we had regretfully omitted in the original submission including the one for PSG-specific expression of *fos-1a* ["To

mark cells of the PSG, we obtained a MosSCI insertion that expresses GFP under a PSG-specific promoter (Sherwood et al., 2005)...”].

What is the reference for saying proximal somatic gonad cells comprise 5% of somatic cells in the adult herm?

This is based on our FACS data (please see specific comment 1 of Reviewer #1). We have changed the wording in the text to “This result is consistent with our estimation from the FACS data that PSG cells comprise only 3-8% of the cell population in the adult *C. elegans* hermaphrodites, ...”

4) Line 171 and Fig. 4: Why isn't wildtype data shown in the figure?

We took the reviewer's suggestion and added the wildtype D1 data to the new Fig. 3b and l (please also see specific comment 4 of Reviewer #1).

5) Line 185 and Fig. 3: Again, why isn't the wildtype data shown in parts b and f of the figure?

We took the reviewer's suggestion and added the wildtype D7 data to the new Fig. 3g and m (please also see specific comment 4 of Reviewer #1).

6) Lines 200-205: Clarify the description of the experiment comparing global *daf-16* and *dos-3* knockdown with knockdown only in the PSGs. For example, reword the sentence to read “...knocking down *dos-3* only in the adult PSGs resembled knocking down *dos-3* in the whole animal, ...” [Bold used only for emphasis.]

We took the reviewer's suggestion and reworded the sentence.

7) Line 229: Substitute “whether” for “no matter”

We took the reviewer's suggestion and added “regardless of” in front of “whether” since we would like to stress that in both self-fertile and mated *C. elegans* hermaphrodites, reduction in Notch signaling contributes to age-related loss of GSPCs.

8) Line 259 and following: Consider including this section with the earlier section starting on line 196 in order to combine the required and sufficient experiments. Fig. 5a could be added to Fig. 3.

We thank the reviewer for this comment. As per Reviewer #1's suggestion (please see major point 1), we performed a series of experiments using the two rescuing *Phsp-16.2::dos-3* cDNA transgenes. Now we include all data generated with these transgenes in the new Fig. 5. Meanwhile, we move D1 GSPC counts of the various genetic groups from the original Fig. S4 to the new Fig. 3 for a better demonstration of the changes in GSPC number during aging.

9) Line 279-280: Substitute “that maintain” for “to” so the line reads “Like other tissue stem cells that maintain the homeostasis of their organ systems, ...”

Here we respectfully disagree with the reviewer. We would like to compare the impact of GSPCs on the reproductive system with that of other tissue stem cells on the organ systems in which they reside. Thus, we prefer to keep the original wording. If the reviewer insists, however, we will make this change.

10) Line 324: I think the authors mean to say "...influences stem cell aging, presumably by functioning as a soluble Notch ligand..."

Yes. We took the review's suggestion and made the change.

11) Figure 1 legend: Describe the volcano plot in terms of the two datasets that are being compared. For example, "b. Volcano plot comparing mRNAs identified by RNA sequencing of D7 ..."

We took the review's suggestion and changed the wording.

12) Figure 2 legend: Clearly state that panel d is a quantification of data represented in panel b.

We took the review's suggestion and added the statement to the figure legend.

13) Figure 3. Part b-e: State how the proliferative zone was identified (parts b-e). By DAPI morphology?

Yes, we use DAPI nuclear morphology to define the proximal boarder of the proliferative zone (the first row of cells in which two or more crescent-shaped meiotic prophase nuclei appear). This information was included in the methods section "Quantification of GSPCs" and is now added to the legend of Fig. 3 as well.

Part f: Has every pairwise comparison been tested for significance? The authors state that *daf-16(-);daf-2(rf)* is not different from *dos-3(-);daf-2(rf)*. However, by eye the difference between those two genotypes looks very similar to the difference between *daf-2(rf)* and *dpy-16(-);daf-2(rf)*, which is indicated as sig different by Student's t-test.

We thank the reviewer for this comment. The point here is that the delay in age-related GSPC loss seen in *daf-2(rf)* is suppressed by removing either *daf-16* or *dos-3*. To facilitate comparison of changes in GSPC counts, we move the D1 data from the original Fig. S4 to the new Fig. 3. Now it is clearly shown in Fig. 3 that the suppressed age-related loss of GSPCs in *daf-2(rf)* is resumed in both *daf-16(0); daf-2(rf)* and *dos-3(0) daf-2(rf)* animals. We clarify this point in the results section "Disruption of *dos-3* function promotes GSPC loss in *daf-2(rf)*" by stating that "Consistent with previous observations, this delay was suppressed by loss of *daf-16*, as evidenced by a reduction in D7 GSPC number in *daf-16(0); daf-2(rf)* animals even though they started with more GSPCs on D1 relative to *daf-2(rf)* (Fig. 3d, i, l, m)" and that "Starting with similar numbers of GSPCs on D1, *dos-3(0) daf-2(rf)* animals maintained significantly fewer GSPCs than *daf-2(rf)* on D7 (Fig. 3e, j, l, m)".

14) Figure 5 legend: Expand on the model description. The illustration suggests DOS-3 is secreted into the pseudocoelom and binds GLP-1 expressed on the distal germ cell plasma membranes. A schematic illustration of the gonad arm might be helpful here.

We took the review's suggestion and added a description of the model to the legend of the new Fig. 6. We also added a schematic illustration of a worm with parts of the gonad arm labeled using a color scheme to the figure.

15) Line 438 and extended data Fig. 2: State in the legend (or elsewhere) what background list of

genes used in the enrichment analysis. Was it the whole genome or a particular subset?

We took the review's suggestion and stated both in the methods section "RNA-seq data analysis" and in the figure legend of Supplementary Fig. 2 that pathway enrichment analysis was performed using Metascape "with the whole genome as the background list".

Reviewer #1 (Remarks to the Author):

The revised manuscript addresses most of the critical points made by both reviewers to make a stronger case. Especially revised Fig. 5 contains important new genetic data to place *dos-3* downstream of *daf-16* and upstream of *glp-1*, in parallel with *lag-2*. This provides strong support for the model presented in Fig. 6. The one point I still believe necessary is to address (my major point 2 in the review of the first version, now also pointed out by reviewer #2 under point 13) regards the method to determine the border of the proliferative zone, which still relies only on DAPI staining. I understand that it is not possible to redo all experiments with antibody staining, but given some of the small effects I would have liked to see a comparison of the two methods at least for one key finding, e.g. quantifying the *daf-2(rf)* and/or *dos-3(lf)* effects at D1 & D7 shown in Fig4 l, m also by HIM-3 staining to determine the border of the proliferative zone and the mitotic index with pH3 staining.

To minor points that may need further explanation:

- Fig 2d now shows a higher magnification of the *dos-3* reporter, as recommended. It is still somewhat difficult to see, but it seems like there is some additional signal outside of the PSG, in the embryos or uterine cells? Please comment.

-The WT controls now included in Fig. 3&4 show a slight but significant decrease in cell number of *daf-2(rf)* at D1 compared to WT- how can this be explained?

Reviewer #2 (Remarks to the Author):

The revised manuscript from Zhang and colleagues, entitled "A non-canonical Notch ligand mediates cell-non-autonomous DAF-16/FOXO activity in antagonizing age-related loss of *C. elegans* germline stem/progenitor cells," has been revised to address my comments. Additional background is provided and many sections are clarified. Additional data requested by reviewer #1 very much strengthen the paper. I still find the wording in what's now line 301-303 to be very awkward ("Like other tissue stem cells to the homeostasis of their organs systems, changes in *C. elegans* GSPCs impact greatly on the functional output..."), but I leave it up to the editor to suggest a revision. Overall, my concerns have been addressed.

We are happy to address additional comments the reviewers raised on our manuscript entitled “A non-canonical Notch ligand mediates cell-non-autonomous DAF-16/FOXO activity in antagonizing age-related loss of *C. elegans* germline stem/progenitor cells”. In our point-by-point response below, reviewer comments are in blue and our responses are in black. Relevant changes in manuscript text are highlighted in yellow.

Reviewer #1 (Remarks to the Author):

The revised manuscript addresses most of the critical points made by both reviewers to make a stronger case. Especially revised Fig. 5 contains important new genetic data to place *dos-3* downstream of *daf-16* and upstream of *glp-1*, in parallel with *lag-2*. This provides strong support for the model presented in Fig. 6. The one point I still believe necessary is to address (my major point 2 in the review of the first version, now also pointed out by reviewer #2 under point 13) regards the method to determine the border of the proliferative zone, which still relies only on DAPI staining. I understand that it is not possible to redo all experiments with antibody staining, but given some of the small effects I would have liked to see a comparison of the two methods at least for one key finding, e.g. quantifying the *daf-2(rf)* and/or *dos-3(lf)* effects at D1 & D7 shown in Fig4 l, m also by HIM-3 staining to determine the border of the proliferative zone and the mitotic index with pH3 staining.

We agree with the reviewer that using a different marker can provide an independent verification of our method and we would like to repeat the experiment as the reviewer suggested. However, we don't have the HIM-3 antibody and it is no longer commercially available. So we use the pS/TQ antibody which also stains the early meiotic prophase nuclei as shown in Yu et al., 2023 to mark the border of the germline proliferative zone. We have analyzed changes in GSPC numbers for all five genetic groups in Fig. 3 using the pS/TQ antibody, and the data are included in the new Supplementary Fig. 4. These data confirm our previous observations made by using DAPI nuclear morphology to determine the border of the proliferative zone: reducing DAF-2 activity delays age-associated depletion of GSPCs and this effect requires *daf-16*, *dos-3*, and *glp-1*.

We have also performed pH3 antibody staining to measure changes in the GSPC proliferation rates. The data are included in the new Supplementary Fig. 4. From these data, we conclude that the mitotic index of all five genotypes analyzed decreases over time, and that none of the mutations seem to have a significant effect on mitotic index.

To minor points that may need further explanation:

- Fig 2d now shows a higher magnification of the *dos-3* reporter, as recommended. It is still somewhat difficult to see, but it seems like there is some additional signal outside of the PSG, in the embryos or uterine cells? Please comment.

We thank the reviewer for this comment. The *dos-3* promoter drives expression in *daf-2(rf)* animals in both spermathecal and uterine cells, just as the *fos-1a* promoter. In our previous paper Qin and Hubbard, 2015, *fos-1a*-expressing cells at the proximal region of the somatic gonad including both spermathecal and uterine cells were collectively referred to as PSG cells. We keep the designation in this manuscript. To clarify, we have specified the term ‘PSG’ in the introduction section of the revised manuscript where it is first mentioned.

-The WT controls now included in Fig. 3&4 show a slight but significant decrease in cell number of *daf-2(rf)* at D1 compared to WT- how can this be explained?

We thank the reviewer for this comment. Insulin signaling promotes germline proliferation during *C. elegans* larval development. Therefore, *daf-2(rf)* animals start with fewer GSPCs on D1. Although it is mostly a cell cycle defect, *daf-2(rf)* animals also have a slightly reduced effective 'reach' of the DTC signal as measured by the distance from the DTC to meiotic entry. These results were published in Michaelson et al., 2010 and our data in Fig. 3&4 are in line with these observations.

Reviewer #2 (Remarks to the Author):

The revised manuscript from Zhang and colleagues, entitled "A non-canonical Notch ligand mediates cell-non-autonomous DAF-16/FOXO activity in antagonizing age-related loss of *C. elegans* germline stem/progenitor cells," has been revised to address my comments. Additional background is provided and many sections are clarified. Additional data requested by reviewer #1 very much strengthen the paper. I still find the wording in what's now line 301-303 to be very awkward ("Like other tissue stem cells to the homeostasis of their organ systems, changes in *C. elegans* GSPCs impact greatly on the functional output..."), but I leave it up to the editor to suggest a revision. Overall, my concerns have been addressed.

Thank you! We took the reviewer's suggestion and changed the wording to "Like other tissue stem cells that maintain the homeostasis of their organ systems, ..."

Reviewer #1 (Remarks to the Author):

The authors did an excellent job in addressing all the remaining points. I think, this version of the manuscript should be accepted for publication.